EMBO
Molecular Medicine

# Oncolytic adenovirus expressing bispecific antibody targets T-cell cytotoxicity in cancer biopsies

Joshua D Freedman[1], Joachim Hagel[1], Eleanor M Scott[1], Ioannis Psallidas[2], Avinash Gupta[3],
Laura Spiers[3], Paul Miller[3], Nikolaos Kanellakis[2] (iD), Rebecca Ashfield[4], Kerry D Fisher[1],
Margaret R Duffy[1] & Leonard W Seymour[1,*] (iD)

## Abstract

Oncolytic viruses exploit the cancer cell phenotype to complete their lytic life cycle, releasing progeny virus to infect nearby cells and repeat the process. We modified the oncolytic group B adenovirus EnAdenotucirev (EnAd) to express a bispecific single-chain antibody, secreted from infected tumour cells into the microenvironment. This bispecific T-cell engager (BiTE) binds to EpCAM on target cells and cross-links them to CD3 on T cells, leading to clustering and activation of both CD4 and CD8 T cells. BiTE transcription can be controlled by the virus major late promoter, limiting expression to cancer cells that are permissive for virus replication. This approach can potentiate the cytotoxicity of EnAd, and we demonstrate using primary pleural effusions and peritoneal malignant ascites that infection of cancer cells with the BiTE-expressing EnAd leads to activation of endogenous T cells to kill endogenous tumour cells despite the immunosuppressive environment. In this way, we have armed EnAd to combine both direct oncolysis and T cell-mediated killing, yielding a potent therapeutic that should be readily transferred into the clinic.

**Keywords** adenovirus; bispecific T-cell engager; BiTE; oncolytic virus
**Subject Categories** Cancer; Genetics, Gene Therapy & Genetic Disease; Immunology

## Introduction

Bispecific T-cell engagers (BiTEs) are fusion proteins generated from single-chain variable fragments (scFv) of two different monoclonal antibodies, one normally a CD3 agonist and the other recognising a surface antigen on target cells, linked by a short flexible peptide linker (Baeuerle & Reinhardt, 2009). By simultaneously engaging CD3 and the target cell antigen, BiTEs mediate T-cell activation and lysis of target cells (Wolf *et al*, 2005). Importantly, BiTE-mediated cytotoxicity can occur without any co-stimulatory signals or *ex vivo* prestimulation (Dreier *et al*, 2002) and independent of peptide-antigen presentation on MHC molecules.

T-cell activation is a result of CD3 clustering on T cells, induced by multiple BiTE molecules simultaneously cross-linking CD3 on T cells and the target cell antigen (Offner *et al*, 2006; Brischwein *et al*, 2007), and leads to increased expression of T-cell activation markers CD69 and CD25, granzyme B, perforin, adhesion molecules and various cytokines such as IFNγ and TNFα. As BiTE-mediated tumour cell killing is not MHC restricted, they provide the opportunity to target T-cell cytotoxicity to tumour cells, independent of antigen recognition and MHC class I levels on the tumour cells.

Blinatumomab, a BiTE targeting CD3 and CD19, is currently the clinically most advanced BiTE (Brischwein *et al*, 2006) and recently gained FDA approval due to its promising clinical performance (Przepiorka *et al*, 2015). Major responses have been demonstrated in various B-cell malignancies (Bargou *et al*, 2008; Topp *et al*, 2011), and BiTEs have been designed targeting different tumour-associated antigens including EpCAM, Her2/neu, EGFR, CEA, EphA2, CD33 and MCSP with some currently under clinical evaluation (Baeuerle & Reinhardt, 2009; Yuraszeck *et al*, 2017).

Oncolytic viruses depend on features of the tumour phenotype to enable productive virus replication, thereby undergoing their lytic life cycle selectively in cancer cells before spreading to infect adjacent cells and repeat the process. Such viruses can also be "armed" to express therapeutic transgenes, and secrete them from infected tumour cells (Seymour & Fisher, 2016). The first agent to receive an FDA product licence, Imlygic, is an oncolytic herpes virus that expresses human GM-CSF (Grigg *et al*, 2016). Several oncolytic viruses are currently under development, but one of the most advanced is EnAdenotucirev (EnAd), a chimera of adenovirus type 3 (Ad3) and type 11p (Ad11p), which is currently undergoing clinical evaluation for a range of metastatic carcinoma types (Kuhn *et al*, 2008; Calvo *et al*, 2014; Illingworth *et al*, 2017).

Here, we assessed whether EnAd could be engineered to express and secrete a BiTE from infected cancer cells that would activate

1   Department of Oncology, University of Oxford, Oxford, UK
2   Oxford Centre for Respiratory Medicine, Oxford University Hospitals NHS Trust, Oxford, UK
3   Churchill Hospital, Oxford University Hospital NHS Trust, Oxford, UK
4   Jenner Institute, University of Oxford, Oxford, UK
    *Corresponding author. Tel: +44 1865 617020; E-mail: len.seymour@oncology.ox.ac.uk

T cells to bind and kill EpCAM-positive target cells. This approach has the advantage of expanding the cytotoxic effects of the virus to kill nearby EpCAM-positive cells, increasing the anticancer potency of the treatment by engaging targeted T-cell cytotoxicity independent of MHCI/TCR engagement. We explored the use of different promoters to regulate BiTE expression and assessed antigen specificity, showing that both CD4[+] and CD8[+] T cells can be activated to mediate cytotoxicity. We also made use of primary human samples of malignant peritoneal ascites and malignant pleural exudates and show that EnAd expressing the EpCAM BiTE can overcome immune-suppressive effects associated with the tumour microenvironment and can activate endogenous T cells to kill endogenous tumour cells. These data make a strong case supporting clinical development of EnAd expressing BiTEs, as a strategy to mediate tumour-targeted oncolysis combined with tumour-targeted immunotherapy, providing a new level of therapeutic strategy for treatment of disseminated cancer.

## Results

### Generation and production of a BiTE targeting EpCAM

A BiTE targeting EpCAM was engineered by joining two scFv specific for CD3ε and EpCAM with a flexible glycine-serine (GS) linker. A control BiTE, recognising CD3ε and an irrelevant antigen (the filamentous hemagglutinin adhesin (FHA) of *Bordetella pertussis*), was also produced. Both BiTEs were engineered to contain an N-terminal signal sequence for mammalian secretion and a C-terminal decahistidine affinity tag for detection and purification (Fig 1A). To characterise the functionality of the recombinant BiTEs, they were cloned into expression vectors under transcriptional control of the CMV immediate early promoter (pSF-CMV-EpCAMBiTE and pSF-CMV-ControlBiTE, respectively). Adherent HEK293 cells (HEK293A) were transfected with the expression vectors and supernatants harvested and concentrated 50-fold for further analysis. To estimate the amount of BiTE produced, samples were serially diluted and evaluated, using anti-His, in a dot blot using decahistidine-tagged cathepsin D as a standard. In this way, it was possible to estimate the level of BiTEs produced into the supernatant to be approximately 20 µg/ml at 48 h post-transfection (of 1.8e7 HEK293A cells)

(Fig EV1A). Specific binding of the EpCAM BiTE and not the control BiTE to recombinant EpCAM protein was demonstrated by ELISA (Fig EV1B).

### Characterisation of human T-cell activation by recombinant EpCAM BiTE

The ability of recombinant EpCAM BiTE protein to activate PBMC-derived T cells was evaluated by adding unstimulated human primary CD3[+] cells to a culture of human DLD colorectal carcinoma cells, which are known to express EpCAM on their surface (Karlsson *et al*, 2008). Addition of 2.5 ng/ml EpCAM BiTE (as supernatant from transduced HEK293A cells) led to a significant increase in T-cell activation markers CD69 and CD25 (Fig 1B), whereas the control BiTE had no effect. Exposure of CD3 cells to the EpCAM BiTE in the absence of tumour cells gave a very modest increase in CD69 and CD25, and this indicates that antibody-mediated clustering of CD3 is essential for full activation by this anti-CD3 binding. Stimulation with the EpCAM BiTE in the presence of tumour cells also induced a significant increase in the generation of gamma interferon-producing T cells (Fig 1C) and cell proliferation (Fig 1D), whereas the control BiTE had no effect. The aim of T-cell activation by BiTEs is to cause degranulation-mediated cytotoxicity, and expression of surface CD107a/LAMP1 (indicating degranulation; Aktas *et al*, 2009) was strongly upregulated by the EpCAM BiTE but not by control (Fig 1E).

The release of cytokines following EpCAM BiTE-mediated activation of PBMC-derived T cells in the presence of DLD cells was characterised by flow cytometry using a cytokine bead array. As before, the control BiTE showed little activity, although the EpCAM BiTE triggered release of several cytokines, including high levels of IL-2, IL-6, IL-10, IL-13, gamma interferon and TNF (Fig 1F). Production of IL-2, gamma interferon and TNF is generally associated with a Th1 response, whereas IL-6 and IL-10 are more often linked to a Th2 response (Mosmann & Sad, 1996).

### Specificity of recombinant EpCAM BiTE

Most human epithelial cells express EpCAM, so to assess whether the effect of the EpCAM BiTE was antigen specific, Chinese hamster ovary cells (CHO cells) were engineered using a lentiviral vector to

---

**Figure 1.  Characterisation of EpCAM BiTE and its effects on PBMC-derived T cells.**

A  Schematic of the structure of the EpCAM-targeted BiTE and non-specific control BiTE. The $V_L$ and $V_H$ domains are connected with flexible peptide linkers (L) rich in serine and glycine for flexibility and solubility. SP, light chain immunoglobulin signal peptide; His, decahistidine affinity tag.

B  Induction of activation markers (i) CD69 and (ii) CD25 on CD3-purified PBMC cultured alone or with DLD cells (5:1) in the presence of BiTE-containing supernatants. CD69 and CD25 were measured by flow cytometry after 24 h of co-culture. Significance was assessed versus IgG isotype.

C  Percentage of IFNγ-positive T cells after 6 h in co-culture with DLD cells (5:1) and BiTE-containing supernatants.

D  Proliferation, represented by division index and percentage of parental T-cell population entering proliferation, of CFSE-stained T cells in co-culture with DLD cells (5:1) and BiTE-containing supernatants. Fluorescence was measured by flow cytometry 5 days after co-culture. Division index was modelled using FlowJo proliferation tool.

E  Degranulation of T cells, measured by CD107a externalisation, in co-culture with DLD cells (5:1) and BiTE-containing supernatants. Externalisation was assessed by co-culture with a CD107a-specific antibody for 6 h followed by flow cytometry analysis.

F  Cytokine levels were measured by LEGENDplex human Th cytokine panel using supernatants from co-cultures of T cells with DLD cells (5:1) in the presence of BiTE-containing supernatants for 48 h.

Data information: Each condition was measured in biological triplicate and data represented as mean ± SD. Significance was assessed versus untreated unless stated otherwise using a one-way ANOVA test with Tukey's *post hoc* analysis, \*P < 0.05, \*\*P < 0.01, \*\*\*P < 0.001.

Source data are available online for this figure.

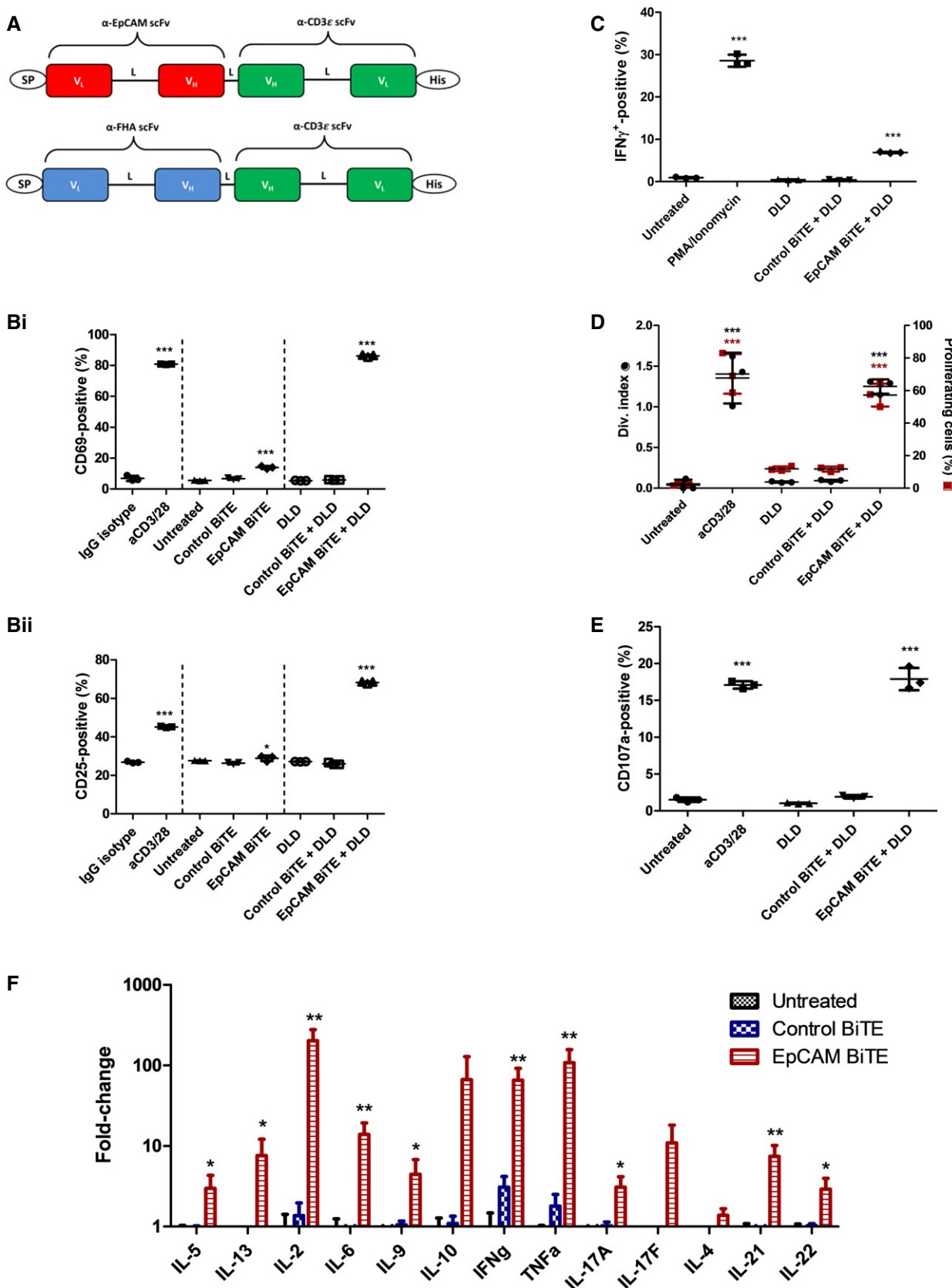

**Figure 1.**

express human EpCAM on their surface. In the presence of EpCAM BiTE and CHO-EpCAM cells, exogenously added PBMC-derived T cells showed strong activation (assessed by CD25 expression Fig 2A) and associated cytotoxicity (Fig 2B) that was not seen with parental CHO control cells or control BiTEs. This indicates that the cytotoxicity of the EpCAM BiTE is antigen specific. We then assessed whether the EpCAM BiTE would kill a range of tumour cells, and whether the level of EpCAM BiTE-mediated cytotoxicity observed was dependent on the density of EpCAM expression. Cytotoxicity of T cells in the presence of the EpCAM BiTE was measured in six different carcinoma cell lines, with greatest cytotoxicity observed in DLD and A431, and least in A549 and PC3 (Fig 2C). This showed a significant association (Pearson's correlation

coefficient, $r = 0.7993$; p=0.0312), with the surface levels of EpCAM (determined by flow cytometry), where A549 and PC3 cells showed the lowest levels and DLD the highest (Fig 2D). This suggests that the presence and level of EpCAM expression do influence the degree of cytotoxicity, although other factors (perhaps the intrinsic resistance of cells to granzyme-mediated apoptosis) also play a role in determining the overall level of cell killing.

## BiTE-mediated activation of CD4[+] and CD8[+] T-cell subsets

To determine which T-cell types are activated by the EpCAM BiTE, PBMC-derived T cells were incubated with DLD cells and activated using the BiTE prior to flow analysis. Both CD4[+] and CD8[+] cells

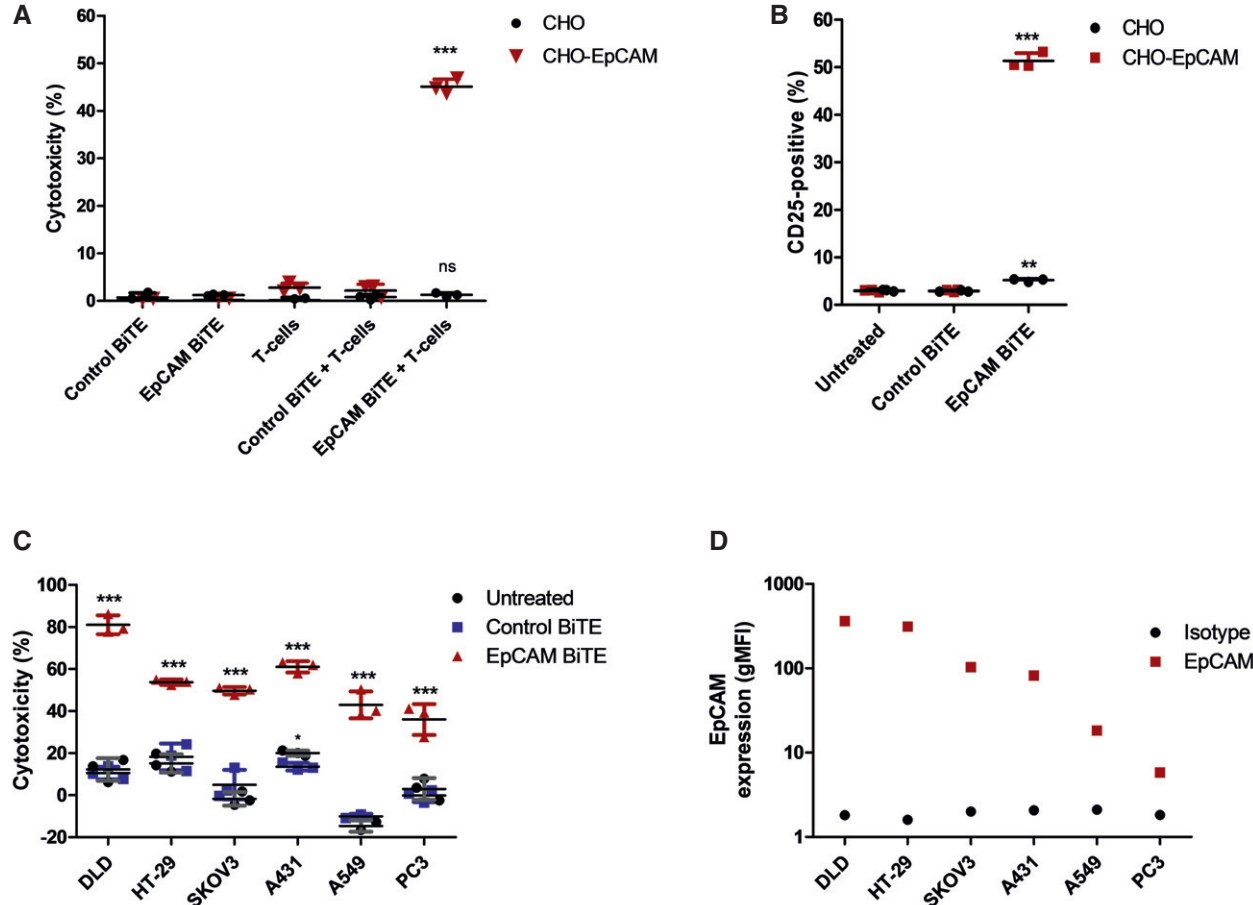

**Figure 2.  Assessment of antigen specificity of EpCAM BiTE-mediated T-cell cytotoxicity.**

A  Induction of activation marker CD25 on CD3[+] T cells in co-culture with CHO or CHO-EpCAM cells (5:1) and BiTE-containing supernatants, measured by FACS analysis after 24 h of co-culture.

B  Cytotoxicity of CHO or CHO-EpCAM cells cultured with BiTE-containing supernatants alone or in co-culture with T cells. Cytotoxicity was assessed by release of LDH into the culture supernatants after 24 h of incubation.

C  Cytotoxicity of multiple EpCAM-positive carcinoma cells after 24 h in co-culture with T cells (1:5) and BiTE-containing supernatants. Viability was measured by MTS assay after 24 h of co-culture.

D  Levels of EpCAM expression ($N = 1$) assessed by FACS analysis of EpCAM-positive cell lines in (C), compared to background fluorescence measured by using an isotype control antibody.

Data information: (A–C) Each condition was measured in biological triplicate and represented as mean ± SD. Significance was assessed versus untreated or T cell-only controls using a one-way ANOVA test with Tukey's *post hoc* analysis, *$P < 0.05$, **$P < 0.01$, ***$P < 0.001$.
Source data are available online for this figure.

showed high levels of expression of CD69 and CD25 (Fig 3A), although the percentage of activated CD4 cells was generally slightly greater. EpCAM BiTE-mediated T-cell proliferation was assessed using CFSE stain (Fig 3B), and degranulation by expression of CD107a/LAMP1 (Fig 3C) and again similar levels of activation were seen for both CD4$^+$ and CD8$^+$ cells. Finally, levels of tumour cell cytotoxicity achieved were compared using EpCAM BiTE to activate purified CD4$^+$ and CD8$^+$ subsets. All T-cell preparations showed similar cytotoxicity (Fig 3D), indicating that both CD4$^+$ and CD8$^+$ cells can contribute to the BiTE-mediated cytotoxicity observed.

**Expression of the EpCAM BiTE from oncolytic adenovirus, EnAdenotucirev**

EnAdenotucirev (EnAd) is an oncolytic adenovirus, a chimera of group B type 11 and type 3 adenovirus with a mosaic E2B region, a nearly complete E3 deletion and a smaller E4 deletion mapped to E4orf4 (Kuhn *et al*, 2008). Currently undergoing several early-phase clinical trials for treatment of cancer, the virus combines good systemic pharmacokinetics and promising clinical activity with the possibility to encode and express transgenes (Calvo *et al*, 2014; Seymour & Fisher, 2016). The EpCAM BiTE was encoded within EnAd immediately downstream of the fibre gene, using a shuttle vector inserted into the virus backbone by Gibson assembly (Fig 4A). The BiTE was placed either under transcriptional control of a CMV immediate early promoter (EnAd-CMV-EpCAM BiTE), or was placed downstream of a splice acceptor site for the adenovirus major late promoter (MLP; EnAd-SA-EpCAM BiTE). In the former configuration, the BiTE should be expressed and secreted whenever the virus successfully infects a cell, whereas expression from the MLP splice acceptor site will occur later on in the infection cycle only when the MLP is activated in cells that are permissive to virus replication. A control BiTE (recognising

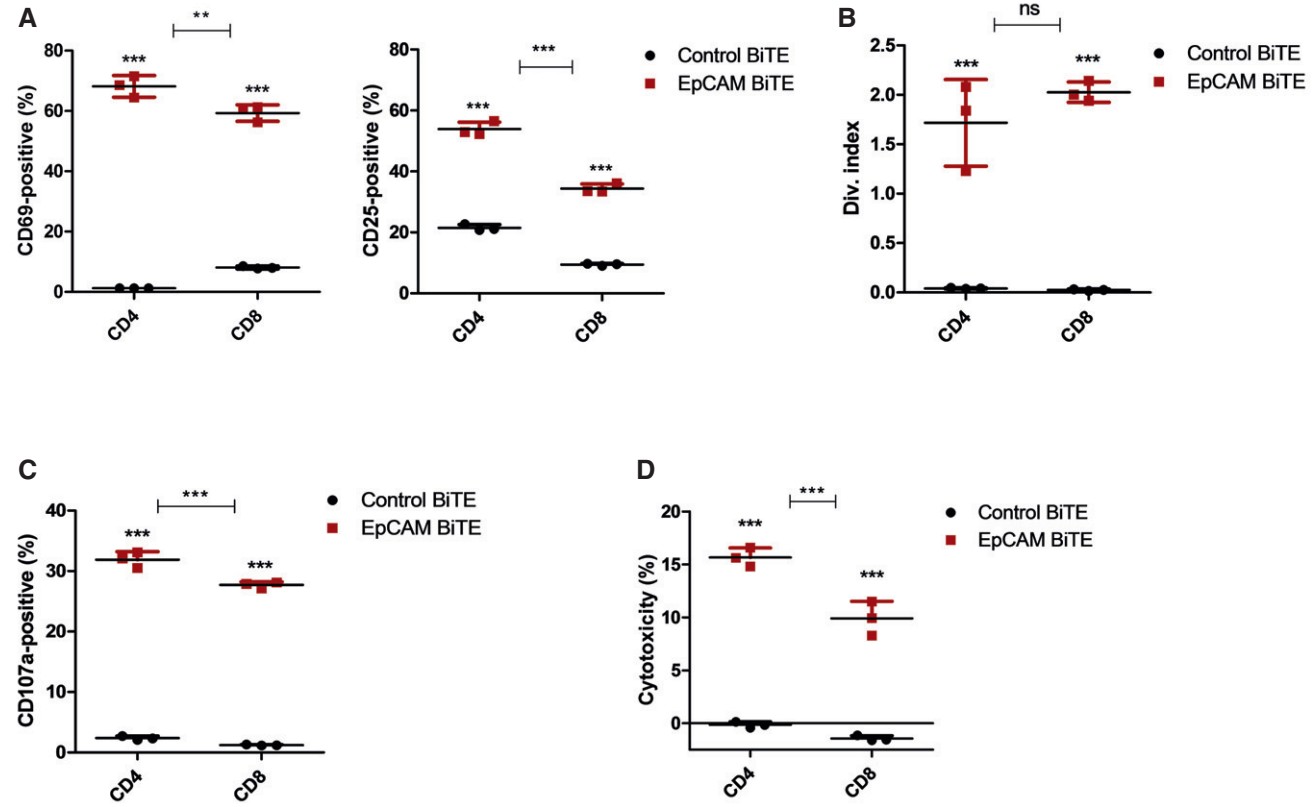

**Figure 3. Identification of which T cells are responsible for BiTE-mediated cytotoxicity.**

A  BiTE-mediated T-cell activation of CD4 and CD8 cells 24 h after co-culture of CD3 T cells with DLD cells (5:1) and BiTE-containing supernatant. Activation was assessed by surface expression of CD69 and CD25 and measured by flow cytometry.

B  Proliferative response of CFSE-stained CD4 and CD8 T cells in co-culture with DLD cells and incubated with BiTE-containing supernatants. Fluorescence was measured after 5-day incubation, by FACS analysis.

C  Degranulation of CD4 and CD8 cells following 6-h co-culture with DLD cells and BiTE-containing supernatants. A CD107a-specific antibody is added to the culture media for the duration of the co-culture, and degranulation is assessed by flow cytometry.

D  Cytotoxicity by either the CD4 or CD8 T-cell subset is assessed by LDH release into supernatant, following 24-h incubation of DLD cells with CD4- or CD8-purified T cells (1:5) and BiTE-containing supernatant.

Data information: Each condition was measured in biological triplicate and represented as mean ± SD. EpCAM BiTE treatment was compared to control BiTE unless stated otherwise, and significance was assessed using a one-way ANOVA test with Tukey's *post hoc* analysis, **$P < 0.01$, ***$P < 0.001$.
Source data are available online for this figure.

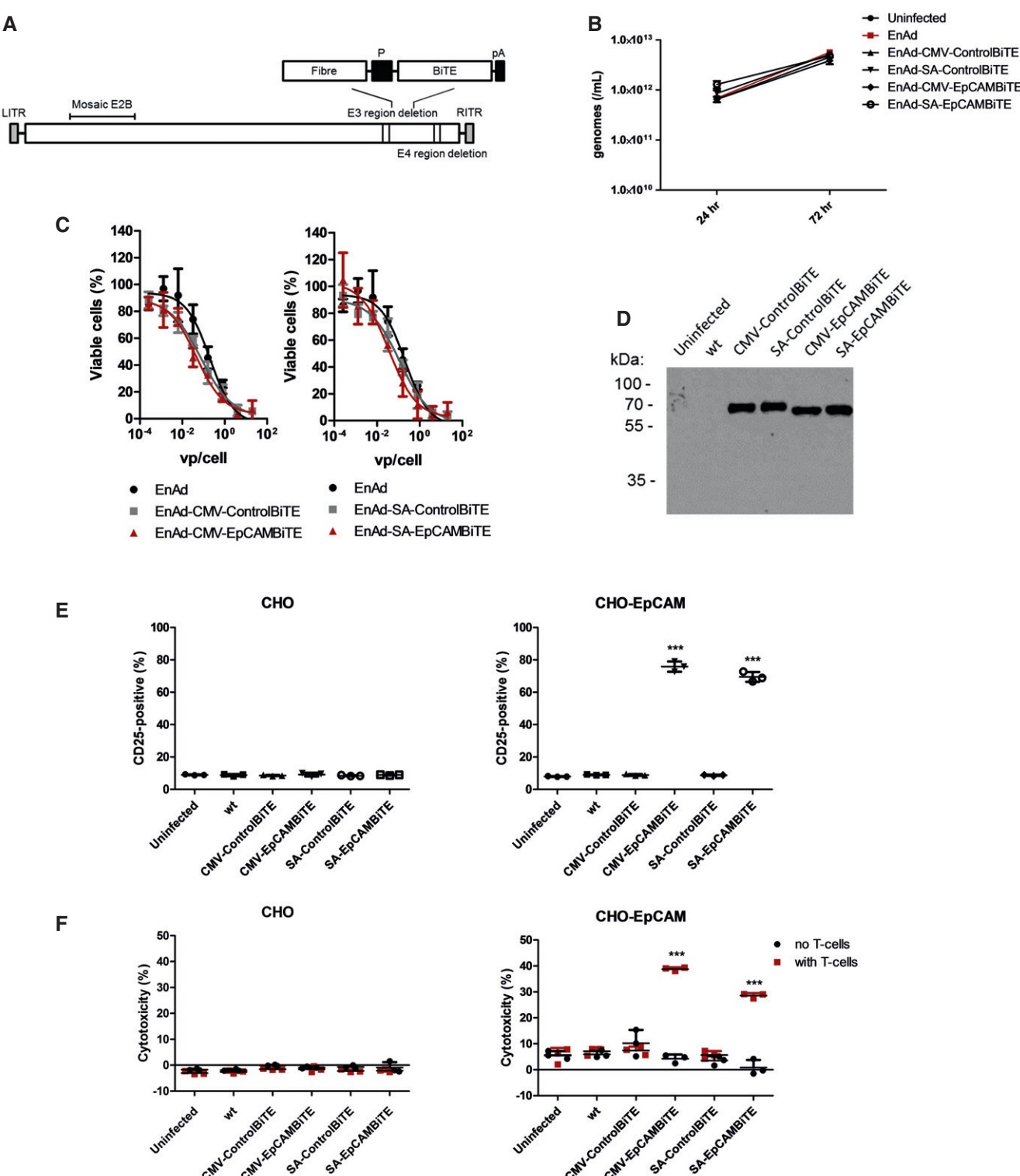

**Figure 4.**

CD3 and FHA) was also introduced to create two corresponding control viruses.

The viruses were cloned, rescued in HEK293A cells, and a large batch of each was prepared in a hyperflask and purified twice by caesium chloride banding, with virus titre and infectious dose of each determined (Fig EV2A). Infection of DLD with parental EnAd and the recombinant BiTE viruses yielded similar amounts of viral genomes (measured by qPCR) at all time points tested, indicating the BiTE transgene does not interfere with the viral replication kinetics (Fig 4B). Next, we investigated the replication and oncolytic

**Figure 4. Characterisation of oncolytic virus EnAd expressing EpCAM BiTE using cell lines and PBMC-derived T cells.**

A   Schematic representation of EnAdenotucirev engineered to express the BiTE transgenes downstream of a CMV promoter or splice acceptor site (P) followed by a polyadenylation sequence (pA).

B   DLD cells were infected with parental EnAd or recombinant virus (100 vp/cell; 2.5e7 vp/ml) and wells harvested at 24 or 72 h. Replication was assessed by measuring genomes using qPCR against viral hexon.

C   Cytotoxicity of DLD cells infected with EnAd or recombinant virus at increasing concentrations of virus. Cytotoxicity was measured by MTS assay after 5 days of infection.

D   Supernatants from day 3 uninfected or virus-infected HEK293A cells were assessed for transgene expression by immunoblot analysis and probed with an anti-His antibody.

E   Induction of activation marker CD25 of CD3-positive T cells cultured with CHO or CHO-EpCAM (E:T 5:1) and diluted HEK293A supernatants from (D). Activation was measured by surface expression of CD25 by flow cytometry.

F   Cytotoxicity of CHO or CHO-EpCAM cells incubated with HEK293A supernatants from (D) alone or in co-culture with CD3-purified PBMC (E:T 5:1). HEK293A supernatants were diluted 300-fold. Cytotoxicity was assessed by LDH released into the supernatant after 24-h incubation.

Data information: Each condition was measured in biological triplicate and represented as mean $\pm$ SD. Significance was assessed using a one-way ANOVA test with Tukey's *post hoc* analysis with each condition compared to untreated, ***$P < 0.001$.

Source data are available online for this figure.

properties of the viruses in the absence of human T cells. DLD cells were infected with virus batches at increasing virus particles (vp)/cell, and the cytotoxicity measured by MTS assay on day 5. All of the recombinant viruses, including those with EpCAM and control BiTEs, regulated by the CMV promoter or splice acceptor, showed cytotoxic activity indistinguishable from the parental virus, showing that the genetic modification had not changed the intrinsic oncolytic activity of the virus (Fig 4C).

To assess BiTE expression and secretion, the BiTE-expressing EnAd viruses were used to infect HEK293A cells, and 72-h supernatants were examined by Western blotting using an anti-His antibody. As shown in Fig 4D, all four viruses (two expressing the control BiTE and two expressing the EpCAM BiTE) showed similar levels of BiTE secreted into the supernatant.

**Selective killing of EpCAM-positive cells by virally produced EpCAM BiTE**

The supernatants from EnAd-EpCAM BiTE-infected HEK293A cells were added to cultures of CHO and CHO-EpCAM cells, either with or without PBMC-derived T cells; T-cell activation and cytotoxicity to the CHO/CHO-EpCAM cells were measured after 24 h. In the case of CHO cells, there was no increase in T-cell expression of CD25 (Fig 4E) nor any cytotoxicity observed with any treatment (Fig 4F). However, T cells incubated with the CHO-EpCAM cells showed substantial increases in CD25 expression using supernatants from HEK293A cells that had been infected with either EnAd-CMV-EpCAM BiTE or EnAd-SA-EpCAM BiTE viruses (Fig 4E). As expected, this translated into selective cytotoxicity to CHO-EpCAM cells only when T cells were added in the presence of supernatant from HEK293A cells that had been infected with either EnAd-CMV-EpCAM BiTE or EnAd-SA-EpCAM BiTE viruses (Fig 4F). Crucially there was no cytotoxicity in the absence of T cells, or when using supernatants from HEK293A that cells had been infected with EnAd expressing the control BiTE.

**Superior cytotoxicity of EnAd expressing EpCAM BiTE**

EnAd kills most carcinoma cells quickly by direct oncolysis (Kuhn *et al*, 2008), although some cells—notably SKOV3 ovarian carcinoma cells—are partially resistant and killed more slowly. We therefore reasoned that the consequences of arming EnAd to secrete

EpCAM BiTE, leading to cytotoxic activation of T cells, might be particularly evident in SKOV3 cells. Cells were therefore exposed to virus (100 vp/cell) 24 h after seeding and cell death monitored by xCELLigence system. PBMC-derived T cells were added (or not) to the SKOV3 cell culture 2 h later. In the absence of T cells, the tumour cells grew for approximately 72 h (manifest by the increasing Cell Index signal in Fig 5A), but cell growth then reached a plateau and remained stable, independent of virus infection, up until at least 160 h. All tested viruses, including parental EnAd, induced no observable target cell cytotoxicity during the time measured. However, when co-cultured with PBMC-derived T cells, both the CMV- and SA-EpCAM BiTE-armed viruses induced rapid SKOV3 lysis, with CMV-driven induced lysis within 16 h, and SA within 44 h following addition of T cells (Fig 5B). Importantly, parental EnAd or the non-specific BiTE control viruses demonstrated no target cell lysis in this time frame even with the addition of T cells. This result was confirmed by LDH assay, in which co-cultures identical to above were set up, with cytotoxicity measured at 24, 48 and 96 h post-infection (Fig EV2B and C). These results are further supported by similar findings in DLD cells in which EpCAM BiTE-expressing viruses induced cytotoxicity at a significantly quicker rate than the control BiTE viruses (Fig EV3A and B). To confirm that target cell cytotoxicity is mediated via T-cell activation, CD3 cells were harvested at each time point and activation status determined by CD69 and CD25 expression, demonstrating similar kinetics of expression as observed for cytotoxicity (Figs 5C and D, and EV3C and D). The approximate quantity of EpCAM BiTE produced from infected DLD cells was determined by comparing cytotoxicity ($Abs_{490}$) induced by infected DLD supernatants to the cytotoxicity induced by known quantities of recombinant BiTE [i.e. creation of a standard curve ($Abs_{490}$)]. DLD cells in co-culture with CD3-purified PBMC (1:5) were incubated with recombinant BiTE (Fig EV3Ei) or supernatants harvested from infected DLD cells (Fig EV3Eii) and LDH release was measured at 24 h. This allowed us to determine that EpCAM BiTE was produced at 165 μg and 50 μg per million DLD for EnAd-CMV-EpCAMBiTE and EnAd-SA-EpCAMBiTE, respectively. The $EC_{50}$ for the EpCAM BiTE is 7.4 ng/ml (Fig EV3E), and therefore, EpCAM BiTE is produced by the recombinant virus at levels that are likely to reach therapeutic doses.

Cytotoxicity of EpCAM BiTE-expressing EnAd was visualised by time-lapse video microscopy. SKOV3 tumour cells (unlabelled) were

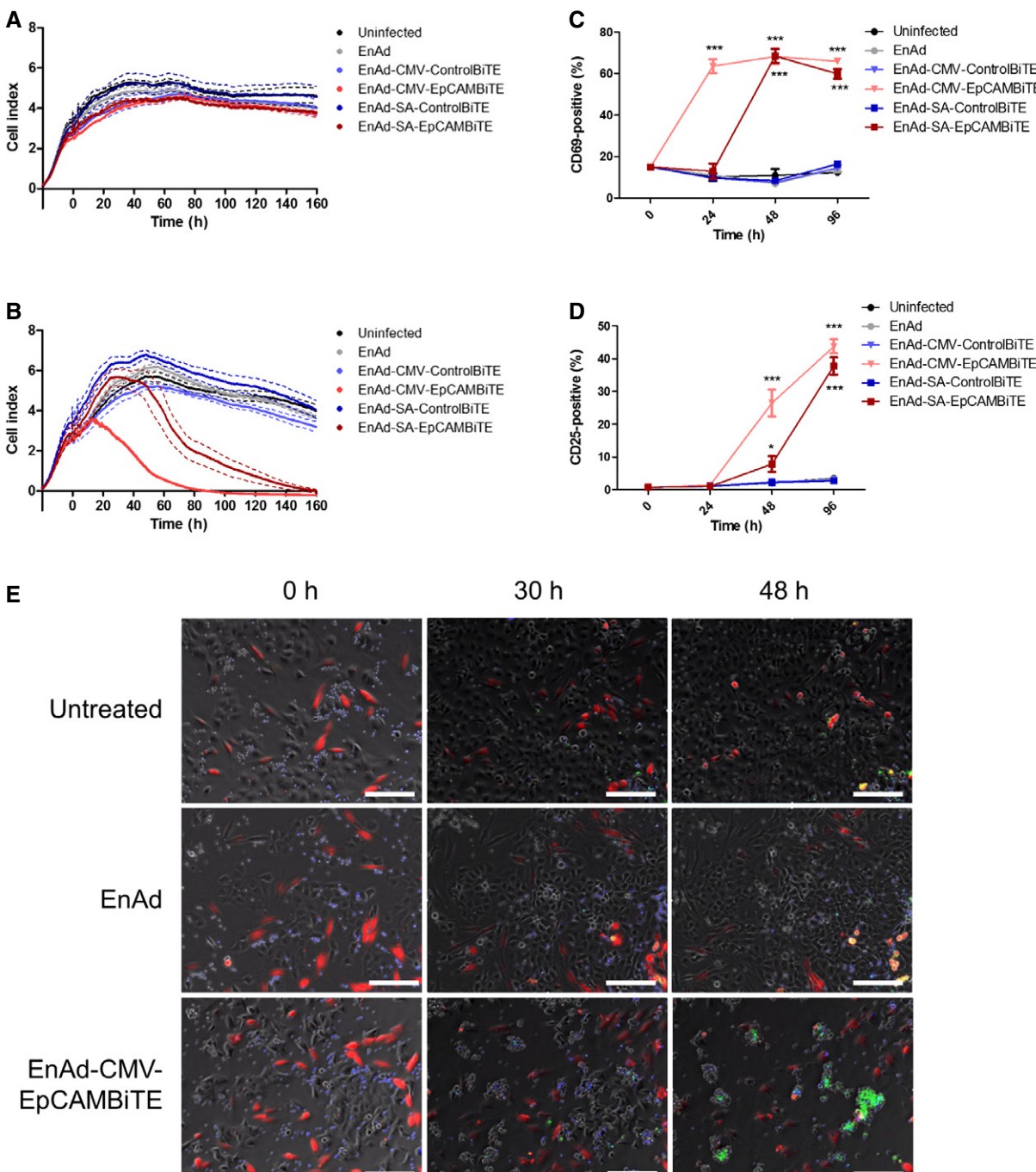

**Figure 5.  Superior potency of EnAd expressing EpCAM BiTE in partially EnAd-resistant cancer cell line.**

A, B    Viability of SKOV3 cells was monitored in real time over 160 h by xCELLigence-based cytotoxicity assay. SKOV3 cells were seeded and infected with EnAd or BiTE-armed EnAd viruses at 0 h, with uninfected cells serving as a negative control. In (B), CD3-purified PBMCs (5:1) were added 2 h post-infection and impedance was measured at 15-min intervals.

C, D    CD3-purified PBMCs were cultured with SKOV3 cells (5:1) that were infected with parental EnAd or recombinant armed viruses. At each time point, T cells were harvested and analysed for surface expression of CD69 (C) or CD25 (D) by flow cytometry.

E    Time-lapse sequences showing co-cultures of SKOV3 carcinoma cells (unstained), NHDF (red) and CD3-purified PBMC (blue), infected with EnAd, EnAd-CMV-EpCAMBiTE or uninfected. Apoptosis was visualised using CellEvent Caspase 3/7 detection reagent (green). Images were taken on a Nikon TE 2000-E Eclipse inverted microscope at intervals of 15 min covering a period of 72 h. Representative images were recorded at the times displayed; original magnification ×10; scale bar, 100 µm. Full time-lapse sequences are displayed in Movies EV1–EV3.

Data information: (A–D) Each condition was measured in biological triplicate and represented as mean ± SD (in A and B, solid line and dotted line, respectively). Significance was assessed by comparison to uninfected control using a one-way ANOVA test with Tukey's *post hoc* analysis, *$P < 0.05$, ***$P < 0.001$.
Source data are available online for this figure.

co-incubated with normal human dermal fibroblasts (NHDF; EpCAM-negative, labelled red, serving as non-target control cells) and PBMC-derived T cells (labelled blue) in the presence of a caspase stain (CellEvent Caspase 3–7 reagent produces a green stain when caspases are activated). Again the combination of EpCAM BiTE-expressing EnAd, combined with exogenous T cells, gave dramatic cytotoxicity to the SKOV3 tumour cells, which showed strong induction of apoptosis when infected with EnAd-CMV-EpCAMBiTE, but not parental EnAd. Importantly, the EpCAM-negative NHDF in co-culture remained viable throughout (Movies EV1–EV3). Representative fluorescent images at different time points from the SKOV3 videos are shown in Fig 5E. Equivalent time-lapse videos showing DLD cells (which are intrinsically more sensitive to the virus) co-cultured with NHDF are also shown (Movies EV4–EV6).

### EpCAM BiTE and EnAd-EpCAMBiTE can overcome immune suppression, activate endogenous T cells and kill endogenous tumour cells within malignant peritoneal ascites

Three clinical samples of malignant peritoneal ascites containing EpCAM-positive tumour cells and primary fibroblasts (as control, non-EpCAM-expressing cells) were expanded *ex vivo* and the mixed primary cell populations were incubated with PBMC-derived T cells and treated with free BiTE or 100 vp/cell EnAd-EpCAMBiTE in culture medium. After 72 h, the level of EpCAM-positive target cells (Fig 6A) or non-target fibroblast activation protein (FAP)-positive fibroblasts (Fig 6B) was measured by flow cytometry. Activation of T cells was analysed by measuring CD25 expression (Fig 6C). Treatment of the samples with free EpCAM BiTE and the EpCAM BiTE-expressing viruses led to strong T-cell

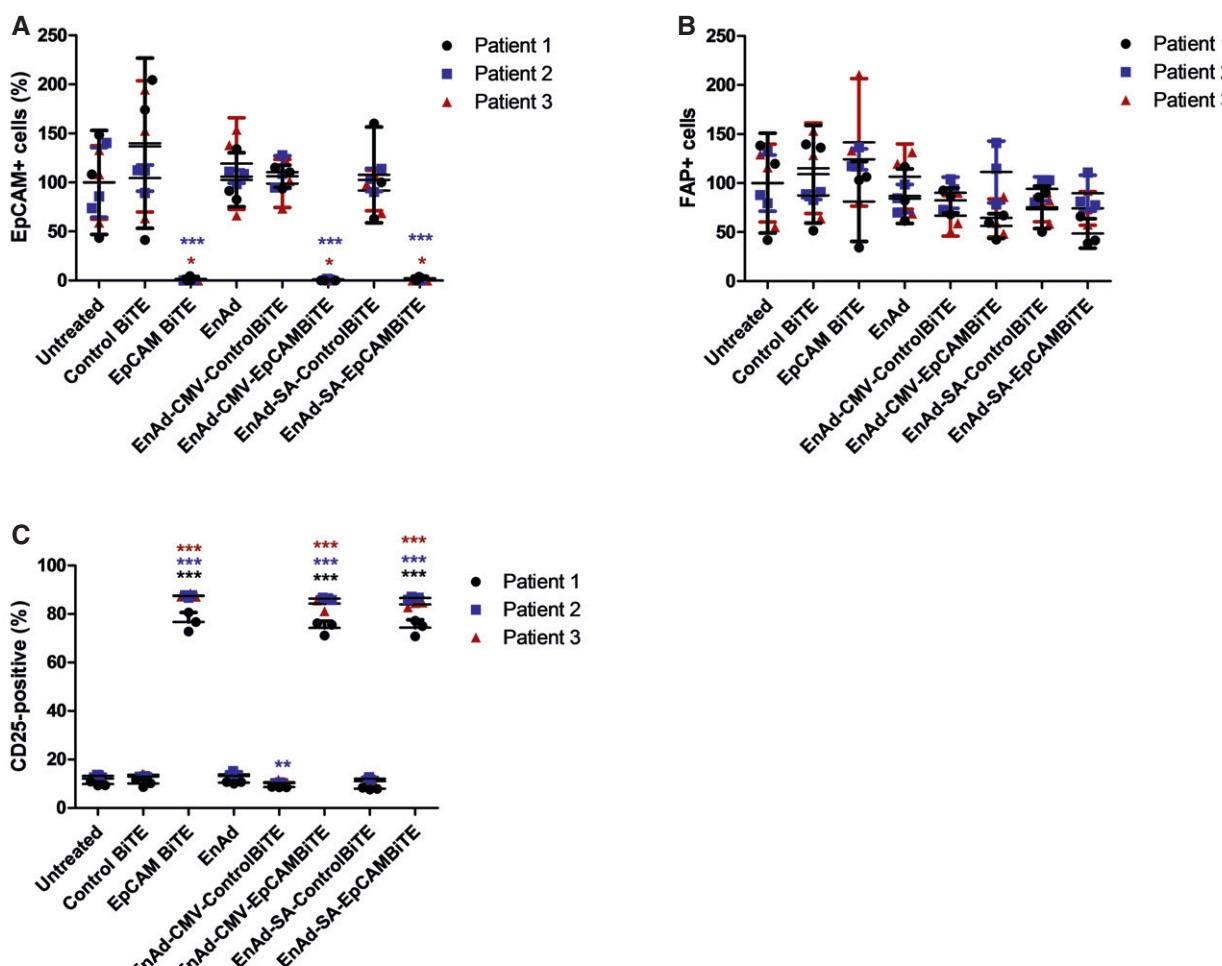

**Figure 6. EnAd expressing EpCAM BiTE can selectively kill primary human tumour cells from chemotherapy-pretreated patients.**

A, B Cytotoxicity of EpCAM+ cells (A) or FAP+ fibroblasts (B), first isolated from three patients' ascites and expanded *ex vivo*, then incubated with recombinant BiTE, or infected with EnAd or recombinant virus. Cytotoxicity was measured by flow cytometry after 5 days.

C Induction of activation marker CD25 on CD3-positive T cells cultured with ascites-derived EpCAM+ and FAP+ cells from (A and B).

Data information: Each condition was measured in biological triplicate and represented as mean ± SD. Significance was assessed by comparison to untreated using a one-way ANOVA test with Tukey's *post hoc* analysis, *$P < 0.05$, **$P < 0.01$, ***$P < 0.001$.
Source data are available online for this figure.

activation (measured by CD25 expression), and a depletion of EpCAM-positive tumour cells to background levels, although FAP-positive (EpCAM-negative) fibroblasts showed no change in numbers. This was observed in all the patients' samples, and none of the other treatments (using the control BiTEs) showed any T-cell activation or cytotoxicity. This demonstrates that the EpCAM BiTE (either free or encoded within an oncolytic virus) can mediate activation of PBMC-derived T cells and selective cytotoxicity to human tumour cells in malignant peritoneal ascites.

Malignant exudates represent an environment of potential immune tolerance with suppressed immune responses commonly observed in patients with late-stage metastatic cancer. The levels of IL-10, considered to be an anti-inflammatory cytokine, were measured in serum from three healthy donors and malignant exudates from 12 patients with peritoneal (A) or pleural (P) malignancies (seven peritoneal ascites—six ovarian and one breast; five pleural—one colorectal, one lymphoma, one lung and two epithelioid mesothelioma). Consistent with the hypothesis that the exudates represent an environment of T-cell immune suppression, IL-10 levels in the exudates (88.1–633.4 pg/ml, mean 375 pg/ml) were far in excess of those measured in normal serum (7.2–10.0 pg/ml, mean 8.9 pg/ml; Fig 7A). The effects of these fluids on T-cell activation was investigated by polyclonally stimulating PBMC-derived T cells with anti-CD3/CD28 beads that mimic *in vivo* T-cell activation in the presence of normal serum, ascites or pleural fluid (all 50%). Whereas in normal serum the anti-CD3/CD28 beads reproducibly gave approximately 60% of T cells dual positive for both CD25 and CD69, the presence of ascites fluid attenuated T-cell activation in 6/12 fluids (Fig 7B). This was strongly correlative with a suppression in T-cell degranulation (Pearson coefficient, $r = 0.8951$; $P < 0.0001$), with CD107a-positive T cells reduced in 10/12 fluids, and was particularly pronounced for patient fluids A1, A2, A7, P3 and P4 (Fig 7C). Importantly, T-cell degranulation negatively correlated with levels of IL-10 in the patient fluids (Pearson coefficient, $r = -0.7645$; $P = 0.0038$). This supports our notion that components of malignant ascites and pleural fluid may exert an immune-suppressive or tolerising effect.

T-cell activation via BiTEs may be able to bypass tumour microenvironment-associated mechanisms of T-cell immunosuppression (Nakamura & Smyth, 2016). We therefore investigated the ability of PBMC-derived T cells and EpCAM BiTE to mediate T-cell activation (CD69, CD25) and degranulation (CD107a) in the presence of normal serum or seven patient malignant exudate samples, of which six were known to be immunosuppressive to T-cell activation by the CD3/CD28 beads (see above). Remarkably, 0/7 fluids attenuated the BiTE-mediated increase in dual-positive CD69$^+$/CD25$^+$ cells, with four actually supporting modest increases in BiTE-mediated T-cell activation compared to normal serum (Fig 7D). Two fluids (A7 and P4) reduced T-cell degranulation (measured by CD107a); however, the decrease induced by A7 was modest (Fig 7E).

Notably, BiTE-mediated T-cell activation was more potent than CD3/CD28 beads, giving higher levels of dual-positive CD69$^+$/CD25$^+$ cells (67.6 vs. 58.9%) and degranulation measured by CD107a (42.9 vs. 22%). Finally, we assessed the ability of BiTEs to activate T-cell cytotoxicity to SKOV3 target cells in the presence of immunosuppressive ascites or pleural fluid following infection with EnAd-SA-control BiTE or EnAd-SA-EpCAMBiTE. PBMC-derived T cells were incubated with normal serum (50%) or four patient fluids (A1, A2, A7, P4) and cytotoxicity evaluated by xCELLigence. All samples showed similar profiles of cytotoxicity to SKOV3 cells, indicating that the virus-encoded BiTEs can overcome the immune-suppressive components of the malignant effusions (including samples A7 and P4) and mediate functional cytotoxicity of T cells (Fig 7F).

In addition to the immune-suppressive fluid and tumour cells present, ascites and pleural effusion samples contain tumour-associated lymphocytes and supporting cells of the tumour stroma, providing a unique tumour-like model system to test BiTE-mediated activation of endogenous patient-derived T cells. The cell types present in patient samples were screened by flow cytometry, with representative FSC-SSC plots illustrated in Fig 8A. Following incubation of total endogenous cells (from patient sample in Fig 8A, effector:target (E:T) ratio 0.49) and ascites fluid with free recombinant BiTE or virus encoding BiTE, activation of endogenous patient T cells was assessed. After 24-h incubation, the EpCAM BiTE (but not the control BiTE) induced CD69 and CD25 expression, similar to that observed in culture medium (Fig 8B). This effect was persistent, and after 5 days, only ascites cells receiving free EpCAM BiTE or virus encoding EpCAM BiTE showed CD25 expression on T cells, with slightly higher levels seen when cells were cultured in the ascites fluid than in RPMI media (Fig 8C). Interestingly, free EpCAM BiTE and EnAd-CMV-EpCAMBiTE mediated proliferation of T cells in whole ascites fluid, and not in basic media, suggesting that basic culture media may lack factors required to support viability or proliferation of

---

**Figure 7. EnAd expressing EpCAM BiTE can overcome immune-suppressive effects of pleural effusion and ascites fluid.**

A    Quantity of human IL-10 in primary malignant exudates. NS, normal serum; A, ascites fluid; P, pleural effusion.

B, C    PBMC-derived T cells were incubated with anti-CD3/CD28 beads in normal serum or exudate fluid (50%) from 12 cancer patients. Induction of T-cell activation markers CD69 and CD25 at 24 h (B) and degranulation (CD107a externalisation) at 6 h (C) were analysed using flow cytometry.

D, E    CD3-purified PBMCs were co-cultured with SKOV3 (5:1) and EpCAM or control BiTE in the presence of normal serum or either pleural effusion or ascites fluid (50%) from seven patients. CD69 and CD25 dual positivity (D) and CD107a externalisation (E) on CD3$^+$ T cells were measured by flow cytometry at 24 and 6 h, respectively.

F    Viability of SKOV3 cells was monitored in real time over 130 h by xCELLigence-based cytotoxicity assay. SKOV3 cells were seeded and incubated with EnAd-SA-ControlBiTE or EnAd-SA-EpCAMBiTE at 16 h. Uninfected cells served as a negative control. CD3-purified PBMCs (5:1) were added 2 h post-infection in the presence of culture medium or either pleural effusion or ascites fluid. Impedance was measured at 10-min intervals.

Data information: (A–E) Each condition was measured in biological triplicate and represented as mean ± SD. Significance was assessed by comparison to normal serum using a one-way ANOVA test with Tukey's *post hoc* analysis. *$P < 0.05$, **$P < 0.01$, ***$P < 0.001$.
Source data are available online for this figure.

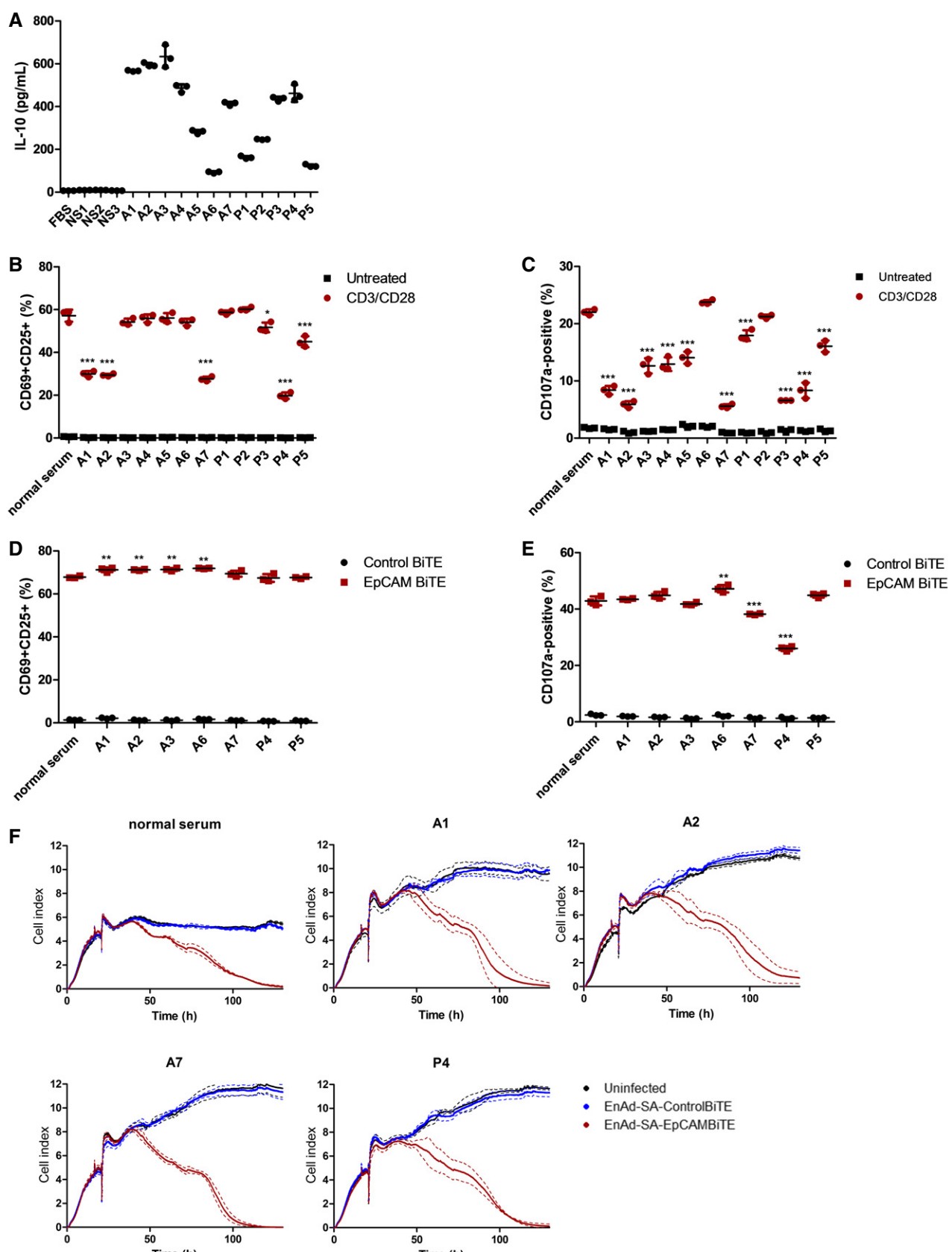

**Figure 7.**

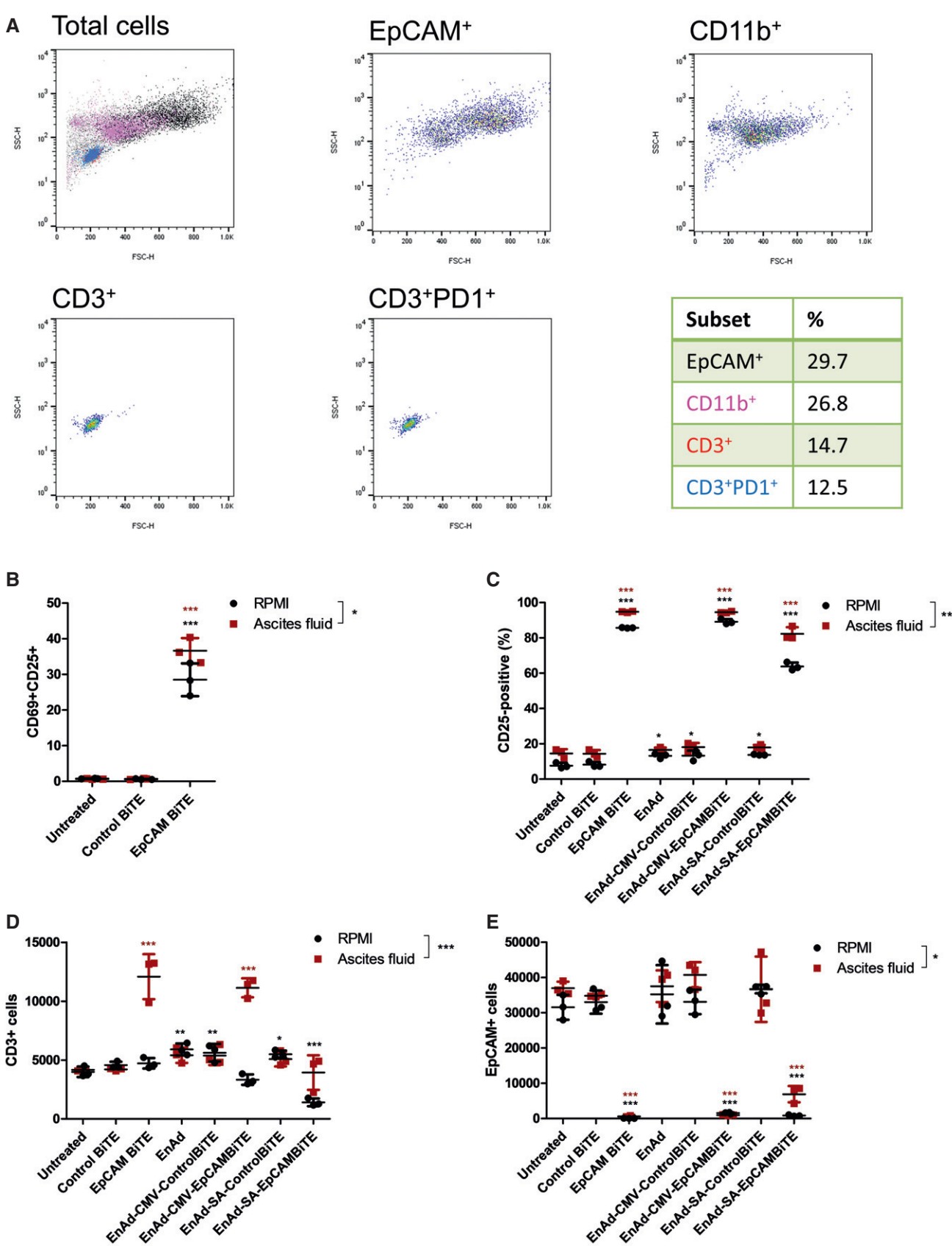

**Figure 8.**

**Figure 8.  EnAd expressing EpCAM BiTE can activate endogenous T cells to kill endogenous tumour cells within culture media and malignant exudate fluid.**

A    Representative spectra (ascites sample, patient 1) demonstrating screening of exudate fluid for its cellular composition, as assessed by flow cytometry. Proportion of each cell type is documented in the table.

B    Endogenous unpurified ascites cells (patient 1) were incubated in RPMI culture medium or 50% ascites fluid in the presence of free EpCAM or control BiTE. After 24 h, the total cell population was harvested, and the number of $CD3^+CD69^+CD25^+$ cells measured by flow cytometry.

C–E    Total ascites cells (patient 1) were incubated in RPMI culture medium or 50% ascites fluid and treated with free BiTE, EnAd or EnAd expressing BiTE. After 5 days, CD25 expression of endogenous T cells (C) and the quantity of $CD3^+$ (D) and $EpCAM^+$ (E) cells were measured by flow cytometry.

Data information: Each condition was measured in biological triplicate and represented as mean ± SD. Significance within each fluid treatment was assessed by comparison to untreated using a one-way ANOVA test with Tukey's *post hoc* analysis. Significance between fluid treatments was assessed by two-way ANOVA test with Bonferroni *post hoc* analysis, *$P < 0.05$, **$P < 0.01$, ***$P < 0.001$.

Source data are available online for this figure.

the ascites T cells (Fig 8D). The absence of an increase in $CD3^+$ cells following EnAd-SA-EpCAMBiTE treatment at this time point may reflect the later onset of expression kinetics relative to the CMV promoter. Cytotoxicity was assessed by measuring residual level of EpCAM cells by flow cytometry. Free EpCAM BiTE and the virus encoding EpCAM BiTE (CMV and SA) caused a similar depletion of EpCAM cells when the experiment was performed in medium and also in 50% ascites fluid (Fig 8E). These data show that the EpCAM BiTE can overcome the immune-suppressive effects of peritoneal ascites or pleural fluid to activate endogenous T cells to kill endogenous tumour cells.

### Identification of malignant exudate cells expressing transgenes

To determine which cells were infected by the EnAd virus, and which cells would produce and secrete BiTEs, we evaluated infection patterns using reporter viruses (EnAd-CMV-GFP and EnAd-SA-GFP). The viruses were added to whole cells isolated from malignant exudates by centrifugation and resuspended in 50% exudate fluid. After 72 h, fluorescence microscopy showed that EnAd-CMV-GFP (which is expected to express GFP transgene in all cells that are infected) showed expression in cells of a range of morphologies (Fig EV4A), whereas EnAd-SA-GFP (which is designed to express GFP transgene only in cells supporting virus replication) was restricted to plump, round cells. Flow cytometry showed that the GFP expression from EnAd-CMV-GFP was found in many macrophages ($CD11b^+$) as well as $EpCAM^+$ cells, whereas EnAd-SA-GFP expression was restricted to $EpCAM^+$ cells (Fig EV4B). This supports the notion that EnAd-CMV-EpCAM BiTE will express BiTEs in a range of cells within the malignant effusions, whereas EnAd-SA-EpCAM BiTE will be restricted mainly to tumour cells.

### EpCAM BiTE and EnAd-EpCAMBiTE show reproducible activity within a range of malignant peritoneal and pleural exudates

To assess inter-patient variability of the EpCAM BiTE-expressing viruses in this highly clinically relevant setting, we obtained several samples of pathological exudates from patients with a range of malignancies. At initial screening, samples considered suitable for further analysis were those containing CD3- and EpCAM-positive cells, having a wide range of E:T ($CD3^+$:$EpCAM^+$) ratios (Fig 9 see legend). We also assessed the expression of PD1 by endogenous T cells following their initial isolation, and whereas only 10% of PBMC-derived T cells express PD1, all the malignant exudate samples T cells were on average 65% positive for PD1 and reached sometimes as high as 93% (Fig EV5).

Total exudate cells (prepared as above) were incubated at fixed concentrations in 50% exudate fluid in the presence of 500 ng/ml free EpCAM BiTE or 100 vp/cell virus encoding BiTE. After 5 days, the total cell population was harvested, and the total number of $CD3^+$ cells (Fig 9A) was measured. Compared to untreated controls, only samples receiving the free EpCAM BiTE or EnAd encoding EpCAM BiTE showed T-cell proliferation. This confirms that the EpCAM BiTE was binding to the EpCAM target and cross-linking CD3 to stimulate endogenous T cells. Treatment with the EpCAM BiTE, EnAd-CMV-EpCAM BiTE and EnAd-SA-EpCAM BiTE typically expanded the T-cell numbers by factors of up to 30-fold. The free and virally encoded EpCAM BiTE also induced significant T-cell activation of tumour-associated lymphocytes (assessed by CD25 expression) in all patients' samples, even within the likely immune-tolerising environment of the malignant exudate fluid (Fig 9B). There was noticeable variation between patients, with activation ranging from 50% to 95% dependent on the donor. This might reflect the broad range of E:T ratios between patients (ranging from

**Figure 9.  EnAd-EpCAMBiTE shows reproducible activity within an expanded cohort of patient malignant peritoneal and pleural exudates.**

A–C    Unpurified total cells from ascites or pleural effusions (from seven different patients; pleural effusion, blue; peritoneal ascites, red) were incubated in 50% fluid from the same exudate sample in the presence of free BiTE, EnAd or recombinant virus. After 5 days, the total cell population was harvested, and the number of (A) $CD3^+$ T cells and those which were (B) $CD25^+$ was quantified. (C) The number of $EpCAM^+$ cells was measured using flow cytometry.

D    Representative microscopy images (magnification ×10; scale bar, 100 μm) and flow cytometry analysis of pleural effusion cells of patient 3 (cancer cells and lymphocytes) following treatment with EnAd or EnAd-SA-EpCAM BiTE. Pink, total cells; blue, $CD3^+$ cells; red, $CD3^+CD25^+$ cells; black, $EpCAM^+$ cells.

E    At 5 days, cytokine levels were measured by LEGENDplex human Th cytokine panel using pleural effusion cultures following incubation with free recombinant BiTE or infection with EnAd or recombinant virus. E:T ($CD3^+$:$EpCAM^+$) ratio upon receipt of patient samples were 0.51 (patient 1), 37.6 (patient 2), 0.39 (patient 3), 4.44 (patient 4), 0.14 (patient 5), 6.12 (patient 6) and 30.6 (patient 7).

Data information: Each condition was measured in biological triplicate and represented as mean ± SD. Significance was assessed by comparison to untreated control samples using a one-way ANOVA test with Tukey's *post hoc* analysis, *$P < 0.05$, **$P < 0.01$, ***$P < 0.001$.

Source data are available online for this figure.

0.13 to 37). Similarly, samples treated with EnAd expressing the EpCAM BiTE showed high activation in some patients (ranging from 10 to 20% up to 95%, for both EnAd-CMV-EpCAM BiTE and EnAd-SA-EpCAM BiTE). Parental EnAd, or EnAd expressing control

BiTEs, or free control BiTEs caused little or no stimulation above background in any sample.

We assessed the ability of the BiTE-expressing viruses to mediate EpCAM-targeted cytotoxicity by measuring residual levels

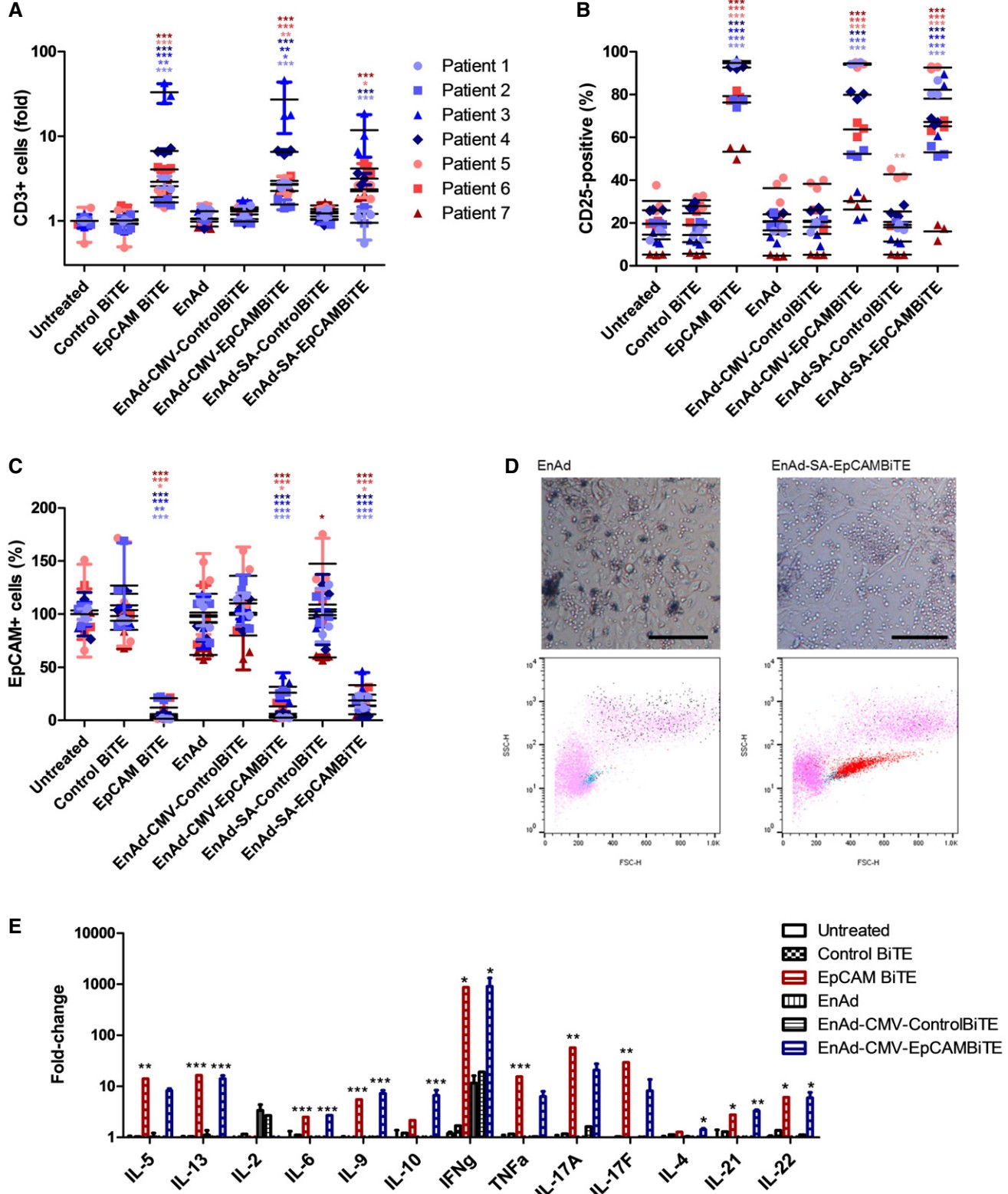

**Figure 9.**

of EpCAM-positive cells by flow cytometry at the end of the 5-day incubation (Fig 9C). The free EpCAM BiTE and the two viruses encoding EpCAM BiTE caused a marked depletion of autologous EpCAM-expressing cells in every case, whereas the other treatments had little or no effect on the level of EpCAM[+] cells. In most patients, EnAd expressing the EpCAM BiTE showed similar levels of T-cell activation and targeted cytotoxicity independent of whether BiTE expression was regulated by the CMV promoter or the SA system. However, one patient (patient 3) showed higher T-cell expression of CD25 using the EnAd-SA-EpCAM BiTE compared to EnAd-CMV-EpCAM BiTE, although the levels of T-cell proliferation and associated cytotoxicity to EpCAM[+] cells were similar for both. The reason for this finding is currently unclear, although it might indicate that even partial T-cell activation is sufficient to eradicate tumour cells in this patient sample.

The different effects of parental EnAd and EnAd-SA-EpCAM BiTE are shown by microscopy in Fig 9D, where expression of the BiTE decreases the presence of tumour cells (seen as large, plump cells) and expands the T-cell population (small round cells). The associated flow cytometry plots confirm the substantial expansion and activation of T cells following treatment with the EpCAM BiTE-expressing virus.

Finally, we characterised the effects of the various treatments by measuring the levels of key cytokines produced using a LEGEN-Dplex protein array (Fig 9E). By far, the greatest fold increases were in gamma interferon, which rose nearly 1,000-fold following treatment with the free EpCAM BiTE or EnAd encoding EpCAM BiTE. These two treatments also caused approximately 10-fold increases in expression of IL-5, IL-13, tumour necrosis factor (TNF), IL17A and IL17F, characteristic of activated T cells. EnAd alone (or expressing the control BiTE) also caused a 10-fold rise in gamma interferon, but otherwise no treatments caused any appreciable changes in cytokine expression.

## Discussion

Oncolytic viruses offer an intriguing new strategy to combine several therapeutic modalities within a single targeted, self-amplifying agent (Keller & Bell, 2016; Seymour & Fisher, 2016). As they replicate selectively within cancer cells and spread from cell to cell, some oncolytic viruses are thought to mediate cell death by non-apoptotic death pathways (Ingemarsdotter *et al*, 2010; Li *et al*, 2013), as part of the process allowing virus particles to escape from dying cells. EnAd, in particular, kills cells by a pro-inflammatory process known as oncosis or ischaemic cell death (Dyer *et al*, 2017). This non-apoptotic death mechanism causes release of several pro-inflammatory cellular components, such as ATP, HMGB1 and exposure of calreticulin (known as damage-associated molecular patterns, DAMPs) (Weerasinghe & Buja, 2012), and is likely pivotal to the ability of the virus to promote an effective anticancer immune response. In addition to the consequences of direct lysis, however, viruses offer the potential to encode and express other anticancer biologics, obviating delivery challenges and ensuring the biologic achieves its highest concentration within the tumour microenvironment. Imlygic encodes GM-CSF; however, the potential for arming viruses is virtually limitless and provides many exciting opportunities to design multimodal therapeutic strategies with additive or synergistic anticancer effects (Hermiston & Kuhn, 2002; de Gruijl *et al*, 2015).

Encoding BiTEs within oncolytic viruses provides a powerful means to activate tumour infiltrating lymphocytes to become cytotoxic and lyse antigen-positive target cells, providing a completely separate therapeutic modality from the effects of direct viral lysis. In this study, we have shown that BiTE-targeted cytotoxicity is fully antigen specific, can be mediated by both CD4 and CD8 T cells (Brischwein *et al*, 2006) and can be incorporated into an oncolytic adenovirus and expressed only in cells that allow virus replication. In addition, the current study shows, for the first time, that endogenous T cells within liquid cancer biopsies can be activated by BiTEs and virus-encoded BiTEs and can kill endogenous tumour cells without any additional stimulation or reversal of immune suppression. Importantly, this can happen even in the primary fluids that comprise the microenvironment of peritoneal ascites or pleural effusions, as surrogates for the immune-suppressive microenvironment of solid tumours.

It is feasible that BiTEs may be able to overcome tumour-associated immune-suppressive effects that limit physiological immune responses, simply by providing excessive stimulation to T cells through simultaneous clustering of multiple CD3 molecules. Since BiTEs employ an antibody structure to recognise cell surface antigens, without the need for HLA presentation, the number of antigens recognised by an individual BiTE may be very large. For example, EpCAM is expressed at levels up to $1 \times 10^6$ copies/cell on malignant exudates (Yao *et al*, 2013). In contrast, most class I HLA molecules are present at < 100,000 copies per cell, and each can contain just one from thousands of different HLA-associated peptides (Schellens *et al*, 2015), meaning that the number of targets for a specific T cell will likely be less than a hundred per cell (Purbhoo *et al*, 2006). T-cell regulation, including checkpoint-mediated suppression, has likely evolved to be active in this range of stimulus—hence, it may be possible for BiTE-mediated recognition of targets to overcome pathophysiological inhibition mechanism simply by excessive stimulatory T-cell signalling. Arming oncolytic viruses to express BiTEs combines two quite distinct therapeutic mechanisms, with the former providing lytic death of tumour cells that are permissive for virus infection, and the latter targeting T-cell cytotoxicity via a specific, chosen, antigen. This provides considerable flexibility in the design of a therapeutic approach, perhaps using the BiTEs to deliver cytotoxicity to tumour-associated cells that are relatively resistant to kill by the virus directly. For example, while we have exemplified the technology here using a BiTE that recognises a carcinoma-associated antigen (EpCAM), it is also possible to use the BiTE approach to target cytotoxicity to tumour-associated fibroblasts or other stromal cells. Indeed, even when the targets for BiTE recognition are not restricted to expression in the tumour microenvironment, by linking BiTE production to virus replication allows expression of the BiTE to be spatially restricted to the tumour, minimising systemic toxicities. This is important, as BiTEs administered intravenously show relatively short circulation kinetics (Klinger *et al*, 2012) and are often associated with considerable on-target off-tumour toxicities (Teachey *et al*, 2013).

The possibility to encode BiTEs within oncolytic viruses has been previously explored using an oncolytic vaccinia virus with an

Ephrin A2-targeting BiTE. This agent showed that the Ephrin BiTE could mediate activation of peripheral blood mononuclear cells (PBMCs) and antigen-targeted killing of tumour cells both *in vitro* and *in vivo*. Intriguingly, although the BiTE could activate T cells, it did not lead to T-cell proliferation without the addition of exogenous IL-2, whereas the BiTE used in the current study led to extensive proliferation both of PBMC *in vitro* and of tumour-associated lymphocytes using the clinical biopsy samples *ex vivo*. Encouragingly, an adenovirus armed with an EGFR-targeted BiTE has also achieved T-cell proliferation *in vitro* in the absence of IL-2 (Fajardo *et al*, 2017). The reason for this difference in behaviour is currently unknown, although it may reflect the different BiTE design, the different oncolytic virus used or perhaps depend on the antigen density giving sufficient cross-linking of CD3 on the T cells.

One central aim of oncolytic virus therapy is to create an anti-cancer T-cell response that recognises patient-specific neoantigens as well as "public" tumour-associated antigens. Lytic viruses may do this by stimulating improved antigen presentation by lysing tumour cells in the context of DAMPs alongside virus-related pathogen-associated molecular patterns (PAMPs). Immunohistochemical staining of resected colon tumours, following intravenous delivery of EnAd, suggests the virus promotes a strong influx of CD8[+] T cells into tumour tissue (R Garcia-Carbonero, E Calvo, R Salazar, I Duran, I Osman-Garcia, L Paz-Ares, JM Bozada, V Boni, C Blanc, L Seymour, S Alvis, B Champion, KD Fisher, Submitted). However, while this is potentially a very powerful approach, adaptive T-cell responses are ultimately dependent on the expression of MHC class I antigens by tumour cells, to allow targeted killing. Loss of MHC expression is a well-documented immune evasion strategy for tumours (Garrido *et al*, 2016). It is noteworthy that both cytotoxic strategies that are immediately engaged by BiTE-armed oncolytic viruses operate independently of MHC class I by the tumour cells, and therefore can be employed to kill cancer cells even when tumour cells have lost MHC expression.

Encoding BiTEs within EnAd provides a particularly promising strategy to achieve targeted expression in disseminated tumours, exploiting the known blood stability and systemic bioavailability of the virus, which has now been studied in several early-phase clinical trials. Notably, in a study where the virus is given intravenously a few days prior to resection of primary colon cancer, subsequent immunohistological assessment of tumour sections showed that the virus had reached to regions through the tumours and gave strong intranuclear hexon signals, indicating successful infection and virus replication selectively in tumour cells. This confirms preclinical data (Di *et al*, 2014; Illingworth *et al*, 2017) indicating that this virus is stable in 100% human blood and should be capable of tumour-targeted infection of disseminated and metastatic malignancies in human patients.

Bispecific T-cell engagers could be encoded by EnAd without any loss of oncolytic virulence (Fig 4C), reflecting the considerable transgene packaging capacity of the virus. The presence of the transgene will not affect the physicochemical properties of the virus particles; hence, the modified viruses should exhibit exactly the same clinical pharmacokinetics as the parental agent, and should be capable of expressing the encoded BiTE selectively within tumours throughout the body. This provides an exciting and potentially very effective new approach to systemically targeted cancer immunotherapy that should now be prioritised for clinical assessment.

# Materials and Methods

### Cell lines

HEK293A, DLD, SKOV3, MCF7, A431, A549, NHDF and PC3 cells (ATCC) were cultured in Dulbecco's modified Eagle's medium (DMEM, Sigma-Aldrich, UK) and CHO cells (ATCC) in Roswell Park Memorial Institute (RPMI-1640, Sigma-Aldrich, UK). All cell lines were authenticated by STR profiling (CRUK Cambridge Institute). Growth media was supplemented with 10% (v/v) foetal bovine serum (FBS, Gibco, UK) and 1% (v/v) penicillin/streptomycin (10 mg/ml, Sigma-Aldrich) and cells maintained in humidified atmosphere at 37°C and 5% $CO_2$. For virus infections and virus plasmid transfections, cells were maintained in DMEM supplemented with 2% FBS. For recombinant BiTE plasmid transfections, cells were maintained in DMEM without FBS. EpCAM expression of target cell lines was determined by flow cytometry. All cell lines were routinely tested for mycoplasma using MycoAlert™ Mycoplasma Detection Kit (Lonza) and were free from contamination during the course of these studies.

### Generation of EpCAM-expressing stable cell lines

The protein sequence of the EpCAM gene (ID: 4072) was obtained from NCBI database and DNA synthesised by Oxford Genetics Ltd (Oxford, UK). The EpCAM gene was cloned into pSF-Lenti vector by standard cloning techniques producing the pSF-Lenti-EpCAM vector. HEK293T cells were transfected using Lipofectamine 2000 with lentivirus EpCAM expression vector alongside pSF-CMV-HIV-Gag-Pol, pSF-CMV-VSV-G, pSF-CMV-HIV-Rev (Oxford Genetics Ltd). Supernatants containing lentivirus were harvested 48 h later and mixed with polybrene (8 µg/ml). Lentivirus/polybrene mixtures were added to CHO cells and incubated at 37°C. On day 4, the supernatant was exchanged for media containing 7.5 µg/ml puromycin. Stable variants were then clonally selected and EpCAM expression of the parental cell lines or stable-transfected variant was determined by antibody staining with EpCAM or isotype control antibody and analysed by flow cytometry. Positive clones were expanded and used in further experiments.

### Preparation of peripheral blood mononuclear cells and T-cell isolation

Peripheral blood mononuclear cells were isolated by density gradient centrifugation (Boyum, 1968) from whole blood leucocyte cones obtained from the NHS Blood and Transplant UK (Oxford, UK). Blood was diluted 1:2 with PBS and layered onto Ficoll (1,079 g/ml, Ficoll-Paque Plus, GE Healthcare) before centrifugation at 400 *g* for 30 min at 22°C with low deceleration. After centrifugation, PBMCs were collected and washed twice with PBS (300 *g* for 10 min at room temperature) and resuspended in RPMI-1640 medium supplemented with 10% FBS. For extraction of CD3-positive T cells from PBMCs, non-CD3 cells were depleted using Pan T Cell Isolation Kit (Miltenyi Biotec, #130-096-535), according

to the manufacturer's protocol. For further isolation of CD4- and CD8-positive T cells, CD3 T cells underwent another round of purification using CD4$^+$ Microbeads (Miltenyi Biotec, #130-045-101).

## Processing primary ascites and pleural effusions

Primary human malignant ascites and pleural effusion samples were received from the Churchill Hospital, Oxford University Hospitals (Oxford, UK) following informed consent from patients with multiple indications of advanced carcinoma, including but not limited to ovarian, pancreatic, breast and lung. This work was approved by the research ethics committee of the Oxford Centre for Histopathology Research (Reference 09/H0606/5+5). Upon receipt, cellular and fluid fractions were separated and fluid used immediately or aliquots stored at −20°C for future analysis. The cellular fraction was treated with red blood cell lysis buffer (Roche, UK) following manufacturer's instructions. Cell number and viability were determined by trypan blue stain. Cell types present in each sample were determined by antibody staining for EpCAM, EGFR, FAP, CD45, CD11b, CD56, CD3, CD4, CD8, PD1 and CTLA4 and analysed by flow cytometry. For *ex vivo* T-cell activation and target cell lysis experiments, fresh cells and fluid were used. In some cases, the adherent cells were passaged in DMEM supplemented with 10% FBS and expanded for later use.

## BiTE engineering and production

Bispecific T-cell engagers were generated by joining two scFvs with specificities against EpCAM or CD3ε with a flexible $(G_4S)_3$ linker. Each scFv was created by the joining of $V_H$ and $V_L$ domains from parental monoclonal antibodies (EpCAM, mAb 5–10; CD3ε, L2K) by a linker. Each BiTE possessed an immunoglobulin light chain (Ig) N-terminal signal sequence for mammalian secretion and a C-terminal decahistidine affinity tag for detection and purification. BiTEs were engineered by standard DNA cloning techniques and inserted into a protein expression vector (pSF-CMV-Amp) for cytomegalovirus (CMV) promoter-driven constitutive protein expression and secretion.

pSF-CMV-EpCAMBiTE or pSF-CMV-ControlBiTE plasmid DNA was transfected into HEK293A cells using polyethylenimine (PEI, linear, MW 25,000, Polysciences, USA) and under the following conditions, 55 μg of plasmid DNA:110 μg PEI (DNA:PEI ratio of 1:2 (w/w)) was added to cells, incubated at 37°C for 4 h, then replaced with fresh serum-free DMEM and further incubated at 37°C, 5% $CO_2$ for 48 h. Cells were transfected in parallel with pSF-CMV-GFP to ensure transfection efficiency. To harvest secreted protein, the supernatant of transfected cells was collected and centrifuged at 350 $g$, 4°C for 5 min to remove cell components. Supernatants were transferred to 10,000 MWCO Amicon Ultra-15 Centrifugal Filter Units (Millipore). After centrifugation at 4,750 $g$ and 4°C, the volume of the retentate was adjusted with the flow through to obtain a 50-fold higher concentration. Aliquots of concentrated protein were stored at −80°C.

## Generation of BiTE-expressing EnAdenotucirev

The plasmids pEnAd2.4-CMV-EpCAMBiTE, pEnAd2.4-SA-EpCAMBiTE, pEnAd2.4-CMV-ControlBiTE, pEnAd2.4-SA-ControlBiTE

were generated by direct insertion of the transgene cassette encoding the EpCAM BiTE or control BiTE into the basic EnAd plasmid pEnAd2.4 using Gibson assembly technology (Gibson *et al*, 2009; Marino *et al*, 2017). The transgene cassette contained a 5′ short splice acceptor sequence or an exogenous CMV promoter, followed downstream by the EpCAM or control BiTE cDNA sequence and a 3′ polyadenylation sequence. A schematic of the inserted transgene cassette is shown in Fig 4A. Correct construction of the plasmid was confirmed by DNA sequencing. The plasmids EnAd2.4-CMV-EpCAMBiTE, pEnAd2.4-SA-EpCAMBiTE, pEnAd2.4-CMV-ControlBiTE and pEnAd2.4-SA-ControlBiTE were linearised by restriction digest with the enzyme AscI prior to transfection in HEK293A cells. The production of virus was monitored by observation of cytopathic effect (CPE) in the cell monolayer. Once extensive CPE was observed the virus was harvested from HEK293A cells by three freeze-thaw cycles. Single virus clones were selected by serially diluting harvested lysate and re-infecting HEK293A cells, and harvesting wells containing single plaques. Serial infections of HEK293A cells were performed once an infection had reached full CPE in order to amplify the virus stocks. Once potent virus stocks were amplified, the viruses were purified by double caesium chloride banding to produce EnAd-CMV-EpCAMBiTE, EnAd-SA-EpCAMBiTE, EnAd-CMV-ControlBiTE, EnAd-SA-ControlBiTE virus stocks. These stocks were titred by TCID50 (50% tissue culture infective dose) and PicoGreen assay (Life Technologies), following manufacturer's instructions.

## Preparation of supernatants

To evaluate BiTE-mediated cytokine release, DLD cells (20,000) were plated with 100,000 CD3$^+$ T cells in 96-well flat-bottom plate alone or with 2 ng/μl EpCAM or control BiTE. After 48-h incubation at 37°C and 5% $CO_2$, supernatants were collected, cell components removed by centrifugation and aliquots stored at −20°C. To assess BiTE transgene expression from recombinant viruses, HEK293A (1e6) or DLD cells (1.2e6) were infected with EnAd-CMV-EpCAMBiTE, EnAd-SA-EpCAMBiTE, EnAd-CMV-ControlBiTE, EnAd-SA-ControlBiTE or EnAd at 100 vp/cell. Cells were cultured for 72 h at which point the cytopathic effect (CPE) was advanced. Supernatants were collected and centrifuged for 5 min, 300 $g$ to remove cell debris and stored at −20°C for future analysis.

## Immunoblotting

Dot blot was used to measure the concentration of recombinant BiTE produced from plasmid transfections. Two-fold serial dilutions of each BiTE and of a protein standard (10×His-tagged (C-terminus) human cathepsin D, Biolegend, #556704) were prepared. The molar concentration of the protein standard was adjusted to represent a BiTE concentration of 100 μg/ml. Two microlitres of each sample and protein standard was directly applied onto a nitrocellulose membrane. The membrane was air-dried, blocked and probed with α-6xHis (C-terminus) antibody (1:5,000, clone 3D5, Invitrogen, UK, #46-0693) for detection of C-terminally His-tagged proteins, followed by washing and incubation with anti-mouse secondary antibody (1:10,000, Dako, #P0161) and detected by application of SuperSignal West Dura Extended Duration Substrate (Thermo Fisher, #34075) according to manufacturer's instructions.

Supernatants of virus-infected HEK293A cells were analysed by Western blotting for BiTE expression. Supernatants were fractionated by SDS–PAGE and transferred to a nitrocellulose membrane according to manufacturer's protocols (Bio-Rad). Membranes were further treated identically to that of dot blot protocol above.

## Enzyme-linked immuno-sorbent assay (ELISA)

To assess EpCAM binding, ELISA plates were prepared by coating overnight at 4°C with human EpCAM/TROP-1 protein (50 ng/well, Sino Biological Inc, #10694-H02H-50). Plates were blocked for 1 h at ambient temperature with 5% BSA, followed by incubation with diluted EpCAM BiTE-, Control BiTE- and empty pSF-CMV vector-transfected HEK293A supernatants (2 h, room temperature). Plates were washed three times with PBS-T and subsequently after every future binding step. Plates were incubated with anti-His (C-term) antibody (1:5,000, clone 3D5, #46-0693, Invitrogen, UK) for 1 h, room temperature, followed by HRP conjugated anti-mouse-Fc (1:1,000 in PBS/5% milk, Dako) for 1 h at room temperature. HRP detection was performed using 3.3.5.5′-tetramethylethylenediamine (TMB, Thermo Fisher), and stop solution was used for terminating the reaction. Absorbance at 450 nm was measured on a Wallac 1420 plate reader (Perkin Elmer).

## Flow cytometry

Flow cytometry analysis was performed on a FACSCalibur flow cytometer (BD Biosciences) and data processed with FlowJo v10.0.7r2 software (TreeStar Inc., USA). A representative example of analysis of flow cytometry data is shown in Appendix Fig S1. For classification of different cellular populations, antibodies specific for CD45 (HI30, Biolegend), CD11b (ICRF44, Biolegend), EpCAM (9C4, Biolegend) and FAP (427819, R&D Systems) were used. For analysis of T-cell populations, the following antibody clones coupled to different fluorophores were used: CD69 (FN50, Biolegend), CD25 (BC96, Biolegend), IFNγ (4S.B3, Biolegend), αCD107a antibody (H4A3, Biolegend), CD3 (HIT3a, Biolegend), CD4 (OKT4, Biolegend), CD8a (HIT8a, Biolegend), PD1 (H4A3, Biolegend). In each case, the appropriate isotype control antibody was used.

## Characterisation of human T-cell activation

### CD69 and CD25 expression levels

The ability of the recombinant EpCAM BiTE or EpCAM BiTE viruses to induce T-cell activation was assessed by surface expression of CD69 and CD25. Human CD3 cells (75,000 cells/well in 96-well flat-bottom plates) from PBMC or ascites samples were cultured alone or with DLD, SKOV3, CHO, CHO-EpCAM or ascites target cells (15,000) in the presence of media alone, EpCAM or control BiTE protein (2 ng/μl) or recombinant virus (100 vp/cell). CD3 cells were incubated with CD3/CD28 Dynabeads (Thermo Fisher, #11131D) as positive control for T-cell activation. Cells were cultured for 24 h at 37°C unless stated otherwise and subsequently harvested with enzyme-free cell-dissociation buffer (Gibco, #13151014). Total cells were stained with antibodies for surface expression of CD69, CD25, CD3, CD4 or CD8 and analysed by flow cytometry. The effect of ascites fluid on T-cell activation (CD69, CD25, CD107a) was investigated by polyclonally activating CD3-purified PBMC (100,000) by

incubating with CD3/CD28 Dynabeads as per manufacturer's instructions) in RPMI-1640 or fluids isolated from the malignant ascites samples.

### IFNγ expression

The ability of the EpCAM BiTE to induce T-cell activity was assessed by IFNγ expression, by co-culture of T cells for 6 h with DLD cells (200,000 CD3 cells/well, 40,000 DLD cells/well in a flat-bottom 96-well plate) and 2 ng/μl recombinant EpCAM or control BiTE. As a positive control, T cells were stimulated with soluble PMA/ionomycin cell activation cocktail (Biolegend, #423301). Brefeldin A (GolgiPlug, BD Biosciences) was added into the culture medium 5 h before harvest, at which point CD3$^+$ T cells were harvested and intracellularly stained for IFNγ expression and analysed by flow cytometry.

### T-cell proliferation

To study T-cell proliferation, 100,000 CFSE-labelled (CellTrace CFSE kit, Invitrogen, #C34554) CD3$^+$ T cells were incubated with 20,000 DLD cells in 96-well plate format, with 2 ng/μl EpCAM or control BiTE. Five days after co-culture, cells were stained for CD3, CD4 or CD8 and CFSE fluorescence of viable CD3$^+$ T cells was measured by flow cytometry, with total cell number normalised using precision counting beads (5,000/well, Biolegend, #424902). Fluorescence data were analysed and modelled using the proliferation function of FlowJo v7.6.5 software. Data are presented as the percentage of original cells that entered a proliferation cycle (%divided) or the average number of cell divisions that a cell in the original population has undergone (Division Index).

### CD107a degranulation

DLD cells (15,000 cells/well) were co-cultured with 75,000 CD3$^+$ T cells in a flat-bottom 96-well plate in the presence of media alone or 2 ng/μl of control or EpCAM BiTE. αCD107a or isotype control antibodies were added directly to the culture medium. Monensin (GolgiStop, BD Biosciences) was added after 1 h of incubation at 37°C and 5% $CO_2$, followed by 5 h of further incubation. Cells were subsequently harvested, stained for CD3, CD4 or CD8 and analysed by flow cytometry.

### Cytokine release

Cytokines within supernatants harvested from cultures of DLD/PBMC or pleural effusion cells were quantified using the LEGENDplex Human T Helper Cytokine panel (Biolegend, #740001) and flow cytometry following the manufacturer's instructions. Cytokines included in the analysis are IL-2, IL-4, IL-5, IL-6, IL-9, IL-10, IL-13, IL-17A, IL-17F, IL-21, IL-22, IFNγ and TNFα.

## In vitro target cell cytotoxicity assay

Target cell cytotoxicity mediated by recombinant BiTE or viruses was assessed by LDH release or MTS assay. Target cells (DLD, SKOV3, HT-29, A431, A549, PC3, CHO, CHO-EpCAM) were co-cultured with CD3, CD4 or CD8 T cells (E:T 5:1) in a flat-bottom 96-well plate in the presence of media alone, diluted supernatants or virus (100 vp/cell). After 24 h of co-culture (unless stated otherwise), supernatants and cells were harvested and cytotoxicity determined by LDH assay (CytoTox 96 Non-Radioactive Cytotoxicity

**The paper explained**

**Problem**
Cancer is a major killer, and oncolytic viruses provide a promising new approach that combines direct toxicity with the potential creation of an anticancer immune response.

**Results**
Here, we have armed an oncolytic adenovirus known as EnAdenotucirev to express a bispecific T-cell engager (BiTE) molecule. The BiTE is secreted from virus-infected tumour cells and activates T cells in the tumour microenvironment to kill chosen target cells. In this case, we targeted them to kill EpCAM-positive tumour cells. BiTEs, and virus-encoded BiTEs, have a remarkable ability to activate a range of T cells to become cytotoxic and can even activate tumour infiltrating lymphocytes (in liquid biopsies form patients) to kill autologous tumour cells.

**Impact**
'Arming' EnAdenotucirev with BiTEs provides an important means to potentiate the anticancer activity of the virus without decreasing its cancer selectivity. Engaging endogenous T cells to kill cancer cells capitalises on the immune-provocative nature of virus infection and allows the immigrant T cells to be used to kill cancer cells directly. These agents are now ripe for assessment in human clinical trials.

Assay, Promega, #G1780) or MTS viability assay (CellTiter 96 Cell Proliferation Assay, Promega, #G3580) as per manufacturer's instructions. Quantity of BiTE produced from virus-infected DLD cells was determined by comparing cytotoxicity induced by diluted viral supernatants to that of a standard curve generated using recombinant BiTE. To evaluate oncolytic activity of the viruses, DLD cells were seeded in 96-well plate (25,000 cells/well) for 18 h at 37°C and 5% $CO_2$, before infection with increasing vp/cell (five-fold serial dilution, 100 to 5.12e-5 vp/cell) or left uninfected. DLD cytotoxicity was measured on day 5 by MTS viability assay. Dose–response curves were fitted and $IC_{50}$ determined using a four-parameter nonlinear fit model integrated into Prism 7 software (GraphPad Software). Cell viability was monitored in real-time using xCELLigence RTCA DP technology (Acea Biosciences). DLD or SKOV3 cells were plated in 48-well E-plate at 12,000 cells/well. Plates were incubated for 18 h, 37°C, 5% $CO_2$, before cells were either treated with BiTE (2 ng/μl) or infected with virus (100 vp/cell) or left untreated. Two hours after infection, 75,000 CD3$^+$ cells were added to the necessary wells. Cell impedance was measured every 15 min for a duration of up to 160 h. For *ex vivo* cytotoxicity assays, unpurified cells from ascites or pleural effusion samples were resuspended in ascites fluid and plated (1.5e5/well) in flat-bottom 96-well plates. After incubation for the stated duration at 37°C, 5% $CO_2$, supernatants were analysed by LDH assay or total cells were harvested by cell-dissociation buffer, stained for CD3, CD25 and EpCAM, and analysed by flow cytometry.

### Viral genome replication and qPCR

The ability of EnAd-CMV-EpCAMBiTE, EnAd-SA-EpCAMBiTE, EnAd-CMV-ControlBiTE, EnAd-SA-ControlBiTE or EnAd to replicate their genomes was analysed by seeding DLD cells in 24-well plate (150,000 cells/well) for 18 h, 37°C, 5% $CO_2$, before infection with 100 vp/cell. Wells were harvested 24 and 72 h post-infection, and DNA purified using PureLink genomic DNA mini kit (Invitrogen, #K182001) according to the manufacturer's protocol. Total viral genomes were quantified by qPCR against EnAd hexon using specific primer-probe set (primers: TACATGCACATCGCCGGA/CGGGCGAACTGCACCA, probe: CCGGACTCAGGTACTCCGAAGCATCCT).

### Microscopy

Bright-field and fluorescence images were captured on a Zeiss Axiovert 25 microscope. Time-lapse videos were obtained to observe viral or T cell-mediated lysis of target cells by EnAd or EnAd-CMV-EpCAMBiTE. Uninfected cells were used as a negative control. NHDF cells were stained with CellTracker Orange CMTMR Dye (Life Technologies, #C2927), and CD3$^+$ cells were stained with CellTrace Violet Cell Proliferation Kit (Life Technologies, #C34557) following manufacturer's protocols. Dyed NHDF were plated in a 24-well plate at 7,500 cells/well in co-culture SKOV3 at 13,500 cell/well. Plates were incubated for 18 h, 37°C, 5% $CO_2$. Cells were then treated with 300 ng/ml EpCAM BiTE or infected with 100 vp/cell of EnAd or EnAd2.4-CMV-EpCAMBiTE or left untreated. After 2-h incubation, 100,000 dyed CD3$^+$ cells were added to necessary wells, in addition to 1.5 μM CellEvent Caspase 3–7 reagent (Life Technologies, #C10423). Images were captured on a Nikon TE 2000-E Eclipse inverted microscope (10× optical objective) at intervals of 15 min covering a period of 60 h or 72 h. Time-lapse videos (12 frames/second) were generated using ImageJ software.

### Statistics

In all cases of more than two experimental conditions being compared, statistical analysis was performed using a one-way ANOVA test with Tukey's *post hoc* analysis or two-way ANOVA test using Bonferroni *post hoc* analysis. All data are presented as mean ± SD. The significant levels used were $P = 0.01$–$0.05$ (*), $0.001$–$0.01$ (**), $0.0001$–$0.001$ (***). All *in vitro* experiments were performed in triplicate, unless stated otherwise.

**Expanded View** for this article is available online.

### Acknowledgements
The authors would like to gratefully acknowledge support from Cancer Research UK (Programme grant #C552/A17720) and the Medical Research Council (doctoral training centre support). M.R.D. is funded by the Kay Kendall Leukaemia Fund (grant number KKL1050). The pEnAd2.4 cloning vector for generating the BiTE-expressing EnAd viruses were kindly provided by PsiOxus Therapeutics Ltd.

### Author contributions
LWS conceived the study. LWS, MRD, KDF and RA designed the study. JDF and JH conducted the experiments. EMS, AG, PM, IP, NK and LS provided primary tumour material. JDF, LWS and MRD wrote the manuscript with input from all the authors.

### Conflict of interest
K.D.F. and L.W.S. own equity or share options in PsiOxus Therapeutics, which is leading clinical development of EnAd and its derivatives.

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
