## [Review Process File · EMBO Molecular Medicine]

Oncolytic adenovirus expressing bispecific antibody targets T-cell cytotoxicity in cancer biopsies

Joshua Freedman, Joachim Hagel, Eleanor M. Scott, Ioannis Psallidas, Avinash Gupta, Laura Spiers, Paul Miller, Nikolaos Kanellakis, Rebecca Ashfield, Kerry D. Fisher, Margaret R. Duffy, Leonard W. Seymour

Corresponding author: Leonard Seymour, University of Oxford

Review timeline:

Submission date:	11 January 2017
Editorial Decision:	16 February 2017
Revision received:	13 April 2017
Editorial Decision:	04 May 2017
Revision received:	15 May 2017

Transaction Report:

Editor: Roberto Buccione

1st Editorial Decision

16 February 2017

Thank you for the submission of your manuscript to EMBO Molecular Medicine. We are sorry that it has taken longer than usual to get back to you on your manuscript. In this case we experienced some difficulties in securing three appropriate expert reviewers, and then obtaining their evaluations in a timely manner. Finally, we also wished to discuss the evaluations further.

As you will see, the reviewers all agree on the overall quality, interest and potential clinical relevance of this study but also, especially reviewers 2 and 3, raise a number of issues.

Reviewer 2, in particular, suggests that in vivo efficacy experimentation is required to support the clinical relevance of the study.

The issue raised by reviewer 2 was further discussed during our reviewer cross-commenting exercise, whereupon there emerged a consensus that although of course in vivo experimentation would be desirable, this would pose significant difficulties and time investment due to the lack of good models to for exploring anti-tumor immune activation by oncolytic (adeno-)viruses. Excluding fully humanized mouse models with autologous tumor/immune grafts, one might consider studies with human xenografts and non-autologous human T cells. However, we agreed that there is still a good rationale and high relevance for the approach as described in the manuscript paper, given that cellular immune responses to endogenous tumor antigens can be studied.

In conclusion, while publication of the paper cannot be considered at this stage, we would be pleased to consider a revised submission, with the understanding that the Reviewers' concerns must be addressed including with additional experimental data where appropriate and that acceptance of

the manuscript will entail a second round of review.

Please note however, we will not be asking you to provide in vivo experimentation unless you have additional data at hand, as long as you carefully deal with the other concerns raised by the reviewers.

Please note that it is EMBO Molecular Medicine policy to allow a single round of revision only and that, therefore, acceptance or rejection of the manuscript will depend on the completeness of your responses included in the next, final version of the manuscript.

As you know, EMBO Molecular Medicine has a "scooping protection" policy, whereby similar findings that are published by others during review or revision are not a criterion for rejection. However, I do ask you to get in touch with us after three months if you have not completed your revision, to update us on the status. Please also contact us as soon as possible if similar work is published elsewhere.

As mentioned above, EMBO Molecular Medicine now requires a complete author checklist (<http://embomolmed.embopress.org/authorguide#editorial3>) to be submitted with all revised manuscripts. Provision of the author checklist is mandatory at revision stage; The checklist is designed to enhance and standardize reporting of key information in research papers and to support reanalysis and repetition of experiments by the community. The list covers key information for figure panels and captions and focuses on statistics, the reporting of reagents, animal models and human subject-derived data, as well as guidance to optimise data accessibility.

We now mandate that all corresponding authors list an ORCID digital identifier. You may do so though our web platform upon submission and the procedure takes < 90 seconds to complete. We also encourage co-authors to supply an ORCID identifier, which will be linked to their name for unambiguous name identification.

Please carefully adhere to our guidelines for authors (<http://embomolmed.embopress.org/authorguide>) to accelerate manuscript processing in case of acceptance.

I look forward to seeing a revised form of your manuscript as soon as possible

***** Reviewer's comments *****

Referee #1 (Remarks):

In this manuscript Freedman and coll. have designed a new bispecific T cell engager (BiTE) binding to EpCAM on cancer cells and to the CD3 receptor on T cells. After assessing its functionality, the authors inserted the EpCAM BiTE into the genome of oncolytic adenovirus EnAdenovucirev (EnAd) under the transcriptional regulation of a heterologous CMV promoter or an endogenous adenovirus late promoter. Cell culture experiments revealed that EpCAM BiTE expressing adenoviruses induce T-cell activation and proliferation thereby engaging a targeted T cell cytotoxicity which reinforces the intrinsic oncotoxicity of the viruses. Notably this enhanced anticancer activity was also demonstrated using primary pleural effusions and peritoneal malignant ascites derived from cancer patients.

Despite the fact that the study may be considered not so novel (BiTEs specific for various antigens including those directed to EpCAM have already been described in the literature with also some examples documented in the context of oncolytic viro-therapy), the strategy indicated has a strong rationale, results shown are interesting (especially those using clinical samples), solid and experimentally sound. The authors convincingly show that BiTEs may be used to extend the anticancer activity of EnAd warranting the clinical translation of such approach.

I have few comments to the authors.

Introduction

Page 2 third paragraph. A more recent reference(s) (in addition to Baeuerle & Reinhardt, 2009) should be provided to document that a number of BiTEs are currently under clinical evaluation.

Page 3 first paragraph. The cancer types for which EnAd is currently tested in clinic should be listed.

Results

Page 8 second paragraph. Which hypothesis? The hypothesis has not been clearly formulated.

Fig. 5 A and B and Fig.7C. XCellIgence results. Figure legend symbols should be illustrated more clearly. Authors may consider to show only the average values with relative standard deviations in order to reduce the number of lines in the figures.

Fig. 7 C and D. Similar experiments should be also carried out using BiTEs expressing EnAds to further confirm virus-mediated activation of T cells in an immunosuppressive environment and appreciate differences in comparison to free BiTEs.

Materials and Methods

Page 15. A reference should be added for "Gibson assembly technology".

Referee #2 (Comments on Novelty/Model System):

Inclusion of an animal efficacy study would raise my enthusiasm for this study.

Referee #2 (Remarks):

This is a very well written and innovative study employing the oncolytic viruses encoding for BiTEs to enhance immunotherapy against cancer. The current study shows, for the first time, that endogenous T cells within liquid cancer biopsies can be activated by BiTEs and virus-encoded BiTEs and can kill endogenous tumor cells. The study is done beautifully, is well controlled, and statistical interpretations are appropriate.

One major concern of this study is the complete absence of any in vivo efficacy data with the constructed approach. This left me very disappointed at the end of a very exciting study.

Minor: Figure 2: The authors state: a loose association with the surface levels of EpCAM and cytotoxicity: The investigators should evaluate if this attains significance with Pearson's correlation.

Referee #3 (Remarks):

Seymour and colleagues report the development of a next generation oncolytic adenovirus, EnAd, encoding an EpCAM- and CD3-binding recombinant antibody (bispecific T cell engager, BiTE) for combined targeting of tumors by viral tumor cell lysis and T cell cytotoxicity. This is a comprehensive study that covers the engineering of a new, gene-encoded BiTE followed by the development of the novel recombinant virus. Both antibody and virus are thoroughly characterized in relevant assays, including several neat multi-cell and multi-read-out experiments. The data is solid and convincing, demonstrating that BiTEs activate both CD4- and CD8-T cells to kill target tumor cells. There is no doubt that the study is of excellent quality.

The manuscript is highly relevant as EnAd is well advanced in clinical translation and - after marketing approval of the first oncolytic virus (T-Vec) - a promising new candidate virus for proceeding into clinical application. While a BiTE-encoding oncolytic vaccinia virus and a therapeutic (thymidine kinase) gene-containing EnAd have been reported before, the essential novelty of the manuscript is the demonstration of virus- and antibody-mediated activity in clinical

material, i.e. ascites and pleural effusion biopsies of cancer patients. Here the authors show that virus-encoded BiTEs can activate both autologous PBMC-derived and endogenous T cells in an immunosuppressive tumor environment resulting in killing of endogenous tumor cells. These results are highly relevant not only because these biopsies closely reflect the in situ situation: Importantly, the biopsies constitute highly desirable fully human test systems for immune responses by oncolytic viruses, for which immunocompetent in vivo models are of limited value due to species-specificity of virus replication. Analysis of oncolytic activity and immune activation in such liquid biopsies might also constitute a test system to screen patients for individuals that are likely to benefit from treatment with the virus therapeutic. Indeed, the authors could report inter-patient variability of T cell activation and tumor cell cytotoxicity in the pleural effusion model.

In this regard the following questions should be addressed by the authors:

- Why was EnAd-BiTE-mediated T cell proliferation and viability of endogenous EpCAM-positive cells not investigated in the ascites model?
- Was there a specific reason to switch to MCF-7 cells when analysing immune suppressive effects of ascites fluid?
- Can the authors comment in what percentage of patients the ex vivo analysis in ascites or pleural effusions will be possible?
- Can the authors show which cells are infected by the oncolytic viruses in the clinical samples? It would be of interest to explore how infection efficiency correlates with T cell activation and how oncolysis contributes to killing of tumor cells.

Minor Comments:

- The authors should provide more information on the BiTE and viruses: From which antibodies were the scFvs derived that were used for generation of BiTEs? How long are the GS linkers? What were the vp/TCID50 ratios of the virus preparations? Was there any difference between the BiTE-encoding and control viruses?
- The authors should show representative diagrams of their FACS analyses of T cells as supplementary figures.
- Why were 293 cells used to show expression and release of BiTEs by EnAd infection? Is cell lysis of tumor cells required for efficient release of BiTEs?
- Fig. 8B: The percentage of CD25-positive cells for patient 3 is much lower for the CMV virus compared with the SA virus. Any explanation?
- Discussion: The authors state to have shown that EnAd-encoded BiTEs are expressed only in cells that allow virus replication. It is not clear on what data this conclusion is based.
- Please refer to Fig. 7E in the results section.
- Typo page 2: CEA

1st Revision - authors' response

13 April 2017

Referee #1

In this manuscript Freedman and coll. have designed a new bispecific T cell engager (BiTE) binding to EpCAM on cancer cells and to the CD3 receptor on T cells. After assessing its functionality, the authors inserted the EpCAM BiTE into the genome of oncolytic adenovirus EnAdenovir (EnAd) under the transcriptional regulation of a heterologous CMV promoter or an endogenous adenovirus late promoter. Cell culture experiments revealed that EpCAM BiTE expressing adenoviruses induce T-cell activation and proliferation thereby engaging a targeted T cell cytotoxicity which reinforces the intrinsic oncotoxicity of the viruses. Notably this enhanced anticancer activity was also demonstrated using primary pleural effusions and peritoneal malignant ascites derived from cancer patients.

Despite the fact that the study may be considered not so novel (BiTEs specific for various antigens including those directed to EpCAM have already been described in the literature with also some examples documented in the context of oncolytic viro-therapy), the strategy indicated has a strong rationale, results shown are interesting (especially those using clinical samples), solid and experimentally sound. The authors convincingly show that BiTEs may be used to extend the anticancer activity of EnAd warranting the clinical translation of such approach.

I have few comments to the authors.

Introduction

Page 2 third paragraph. A more recent reference(s) (in addition to Baeuerle & Reinhardt, 2009) should be provided to document that a number of BiTEs are currently under clinical evaluation.

This has been attended to – the reference (Yuraszeck, 2017) has been added.

Page 3 first paragraph. The cancer types for which EnAd is currently tested in clinic should be listed.

The virus is being trialled against metastatic carcinomas of any origin. The text has been modified to specify this.

Results

Page 8 second paragraph. Which hypothesis? The hypothesis has not been clearly formulated.

This has now been clarified in the revised text.

Fig. 5 A and B and Fig.7C. XCellIgence results. Figure legend symbols should be illustrated more clearly. Authors may consider to show only the average values with relative standard deviations in order to reduce the number of lines in the figures.

The legend and symbols have been clarified, as suggested, to facilitate the interpretation of the graphs. The dotted lines on either side of the solid lines show the standard deviations of the data. The key point from 5A is that the lines are pretty much superimposed (i.e. no cytotoxicity is induced by the virus in the absence of T cells), hence it makes little sense to show these lines on separate graphs. The key points of B are clear from the graph – namely that virus encoding BiTEs show good cytotoxicity in the presence of T cells, but all the other treatments do not. We hope the clarified legend and symbols now makes this easy to interpret.

Fig. 7 C and D. Similar experiments should be also carried out using BiTEs expressing EnAds to further confirm virus-mediated activation of T cells in an immunosuppressive environment and appreciate differences in comparison to free BiTEs.

Figure 7 has been completely replaced, in line with these comments, and additional data added (new Figure 7 and Figure 8). We have expanded the xCELLigence study (previous 7C) to focus on activation of PBMC T cells by virus-encoded BiTEs in the presence of suppressive fluids, showing that virus-encoded BiTEs can overcome the immune inhibition (now Figure 7F). We have also compared free BiTEs and virus-encoded BiTEs to activate endogenous T cells in tumour ascites (Figure 8C-E) and show that both formats can overcome suppressive effects of ascites fluid. The text has been revised to included description and discussion of these new data.

Materials and Methods

Page 15. A reference should be added for "Gibson assembly technology".

This reference is now included as requested.

Referee #2

Comments on Novelty/Model System

Inclusion of an animal efficacy study would raise my enthusiasm for this study.

These agents are not cross-reactive with animal models (neither the BiTEs nor the oncolytic viruses) hence we appreciate the Editorial guidance that an animal study will not be essential provided all other aspects of our response are strong.

Remarks

This is a very well written and innovative study employing the oncolytic viruses encoding for Bites to enhance immunotherapy against cancer. The current study shows, for the first time, that endogenous T cells within liquid cancer biopsies can be activated by BiTEs and virus-encoded BiTEs and can kill endogenous tumor cells. The study is done beautifully, is well controlled, and statistical interpretations are appropriate.

One major concern of this study is the complete absence of any in vivo efficacy data with the constructed approach. This left me very disappointed at the end of a very exciting study.

Please see above.

Minor: Figure 2: The authors state: a loose association with the surface levels of EpCAM and cytotoxicity. The investigators should evaluate if this attains significance with Pearson's correlation.

We have analysed the data on levels of EpCAM expression (Figure 2D) and cytotoxicity (Figure 2C) and show a Pearson correlation (0.7993) and significant association (p=0.0312). This information is now included in the revised text.

Referee #3

Seymour and colleagues report the development of a next generation oncolytic adenovirus, EnAd, encoding an EpCAM- and CD3-binding recombinant antibody (bispecific T cell engager, BiTE) for combined targeting of tumors by viral tumor cell lysis and T cell cytotoxicity. This is a comprehensive study that covers the engineering of a new, gene-encoded BiTE followed by the development of the novel recombinant virus. Both antibody and virus are thoroughly characterized in relevant assays, including several neat multi-cell and multi-read-out experiments. The data is solid and convincing, demonstrating that BiTEs activate both CD4- and CD8-T cells to kill target tumor cells. There is no doubt that the study is of excellent quality.

The manuscript is highly relevant as EnAd is well advanced in clinical translation and - after marketing approval of the first oncolytic virus (T-Vec) - a promising new candidate virus for proceeding into clinical application. While a BiTE-encoding oncolytic vaccinia virus and a therapeutic (thymidine kinase) gene-containing EnAd have been reported before, the essential novelty of the manuscript is the demonstration of virus- and antibody-mediated activity in clinical material, i.e. ascites and pleural effusion biopsies of cancer patients. Here the authors show that virus-encoded BiTEs can activate both autologous PBMC-derived and endogenous T cells in an immunosuppressive tumor environment resulting in killing of endogenous tumor cells. These results are highly relevant not only because these biopsies closely reflect the in situ situation: Importantly, the biopsies constitute highly desirable fully human test systems for immune responses by oncolytic viruses, for which immunocompetent in vivo models are of limited value due to species-specificity of virus replication. Analysis of oncolytic activity and immune activation in such liquid biopsies might also constitute a test system to screen patients for individuals that are likely to benefit from treatment with the virus therapeutic. Indeed, the authors could report inter-patient variability of T cell activation and tumor cell cytotoxicity in the pleural effusion model.

In this regard the following questions should be addressed by the authors:

- Why was EnAd-BiTE-mediated T cell proliferation and viability of endogenous EpCAM-positive cells not investigated in the ascites model?

These data have been expanded and new experiments added to the revised text, and are shown in the modified Figure 7 (studying activation of PBMC in ascites fluid) and new Figure 8 (showing activation of endogenous T cells in whole ascites samples). Virus-encoded BiTEs

mediated T cell activation (measured by CD25) for both PBMC incubated in ascites fluid (Fig 7F) and also endogenous T cells in whole ascites (Fig 8C). The EnAd-BiTE (and free EpCAM BiTE) mediate significant decrease in the level of EpCAM-positive cells, as expected (Fig 8E).

- Was there a specific reason to switch to MCF-7 cells when analysing immune suppressive effects of ascites fluid?

This is a good point - accordingly we have repeated the study using SKOV3 cells, and those data are now shown instead (Figure 7F)

- Can the authors comment in what percentage of patients the ex vivo analysis in ascites or pleural effusions will be possible?

This is a multifactorial question that depends on hospital practice etc. However incidence of ascites increases with disease progression of ovarian cancer, and 17% of early stages (I and II) and up to 89% with advanced ovarian cancer (III and IV) show significant peritoneal ascites, and could in principle be suitable for analysis (Shen-Gunther, 2002). We considered carefully whether it would be helpful to introduce this information into the manuscript, and eventually decided not - because it would give the impression we are proposing to use this as a biomarker to guide treatment. At this stage that is not our purpose, we are simply using patient samples to inform on the likely activity of this therapeutic approach, and I believe it is premature to raise considerations of precision medicine.

- Can the authors show which cells are infected by the oncolytic viruses in the clinical samples? It would be of interest to explore how infection efficiency correlates with T cell activation and how oncolysis contributes to killing of tumor cells.

This was assessed using two GFP-reporter EnAd viruses. The first has reporter GFP controlled by the CMV promoter (which reports on simple infection of cells) and the second has GFP expression linked to virus replication (indicating permissivity of the cell to virus replication). Using these systems the CMV-GFP virus showed infection of both EpCAM+ and CD11b+ cells, whereas virus replication (using EnAd SA-GFP) was restricted to EpCAM+ cells (Figure EV5B). This suggests that virus replication is restricted to tumour cells, as expected, and is now discussed in the revised text (Expanded View Figure 4).

Minor Comments:

- The authors should provide more information on the BiTE and viruses: From which antibodies were the scFvs derived that were used for generation of BiTEs? How long are the GS linkers? What were the vp/TCID50 ratios of the virus preparations?

The EpCAM scFvs is derived from a monoclonal antibody called mAb 5-10, and CD3ε from a humanised derivation of a monoclonal antibody called L2K. The linkers used are (GGGGS)₃. This has been added to the text in the Materials & Methods section. The requested vp/TCID50 ratios for the virus preparations are below:

	Virus titre (vp/mL)	Infectious particles (PFU/mL)	Ratio
EnAd	4.64E+12	2.91E+11	15.9
EnAd-CMV- ControlBiTE	4.81E+11	1.00E+10	48.1
EnAd-CMV- EpCAMBiTE	1.00E+12	2.51E+10	39.9
EnAd-SA-ControlBiTE	1.61E+12	3.98E+10	40.6
EnAd-SA-EpCAMBiTE	1.22E+12	3.16E+10	38.7

This information has been added to the Materials and Methods and Expanded View Figure 2A.

Was there any difference between the BiTE-encoding and control viruses?

There appears to be slightly reduced virus yields and increased ratios for recombinant virus (above) compared to parental EnAd. However, there were no appreciable differences in oncolytic activity (Figure 4C) or virus replication (Figure 4B) between BiTE-encoding and parental control virus.

- The authors should show representative diagrams of their FACS analyses of T cells as supplementary figures.

This has now been included as requested (Supplementary Figure 1)

- Why were 293 cells used to show expression and release of BiTEs by EnAd infection? Is cell lysis of tumor cells required for efficient release of BiTEs?

293 cells are used routinely in our lab for these studies, although the virus does not require E1A in trans to proliferate. Release of the BiTE results from a signal peptide (active secretion) and is independent of cell lysis. This is now explained more clearly in the revised text.

- Fig. 8B: The percentage of CD25-positive cells for patient 3 is much lower for the CMV virus compared with the SA virus. Any explanation?

A possible explanation for this finding is that in tumour cells from patient 3 the SA promoter operates more effectively than the CMV promoter. We know this occurs in cells that are highly permissive for virus replication. Interestingly, the levels of T cell proliferation and associated cytotoxicity to EpCAM⁺ cells were similar for both viruses, indicating that even partial T cell activation is sufficient to eradicate tumour cells in this patient sample. We have added a sentence to the revised text to explain this possibility. We have also included data from three additional patients (ascites samples) to assess whether we see this effect in other patients (Figure 9A). We have not seen it again, hence we believe it is a relatively rare occurrence.

- Discussion: The authors state to have shown that EnAd-encoded BiTEs are expressed only in cells that allow virus replication. It is not clear on what data this conclusion is based.

This follows from the design of the BiTE expression construct, since the BiTE will only be expressed when the virus major late promoter is activated. These data are further supported by the new Expanded View Figure 4.

- Please refer to Fig. 7E in the results section.

Figure 7E has been replaced, and is now referred to in the text.

- Typo page 2: CEA

This has been attended to.

2nd Editorial Decision

04 May 2017

Thank you for the submission of your revised manuscript to EMBO Molecular Medicine. We have now received the enclosed reports from the referees that were asked to re-assess it. As you will see the reviewers are now globally supportive and I am pleased to inform you that we will be able to accept your manuscript pending the following final amendments:

- 1) Please address the final points indicated by reviewer 3
- 2) Movies cannot be referenced from EV figures. Please rename them as Movie EV1, Movie EV2, etc. Also, the legends should be removed from the main manuscript and zipped together with the

movie file. Please amend EV figure legends after removal of movie links. Finally, please make sure the callouts in the manuscript are updated.

3) Please remove the Fig1S legend from the manuscript and combine with Fig. S1 I the appendix. Please use Appendix Figure S1 nomenclature.

4) Please update references to the 20 Names et al. format

5) Please remove "Expanded View" prefix from the figure callouts.

6) Please correct Fig. EC3F callout I the manuscript

7) Please make scale bars more visible (contrasted colour) in Fig. 5E and add to Fig EV4A

8) Please remove formatting tracking information, as it is no longer needed at this stage.

9) We encourage the publication of source data, with the aim of making primary data more accessible and transparent to the reader. Would you be willing to provide a PDF file per figure that contains the original, uncropped and unprocessed scans of all or at least the key gels used in the manuscript and/or source data sets for relevant graphs? The files should be labeled with the appropriate figure/panel number, and in the case of gels, should have molecular weight markers; further annotation may be useful but is not essential. The files will be published online with the article as supplementary "Source Data" files. If you have any questions regarding this just contact me.

Please submit your revised manuscript within two weeks. I look forward to seeing a revised form of your manuscript as soon as possible.

***** Reviewer's comments *****

Referee #1 (Remarks):

All my concerns have been addressed satisfactorily.

Referee #2 (Remarks):

The authors have answered all previous queries.

Referee #3 (Remarks):

The manuscript of Seymour and colleagues has further and considerably improved during revision, including the addition of results of extensive new experimentation using patient-derived material. The new data substantiates the authors' conclusion that a BiTE-encoding oncolytic EnAd virus shows potent virus- and antibody-mediated anti-cancer activity. Thus, the study now demonstrates EnAd-BiTE-mediated activation of endogenous T cells and killing of endogenous EpCAM-positive cells in both of their established and highly relevant models of ascites as well as pleural effusion biopsies from a large panel of cancer patients. All criticism raised has been adequately addressed in the revised manuscript.

Minor formal points that should be addressed in a final version of the manuscript:

- In Materials & Methods, Cell lines: The method used for testing mycoplasma contamination is missing.
- Legend Fig. 9: The authors should mention that samples patient 1 to 4 are from pleural effusions and patient 5 - 7 from ascites.
- New data shown in Fig. 8D: The authors should mention and comment on the surprising difference between the results for EnAd-SA-EpCAMBITE versus EnAd-CMV-EpCAMBITE.

1) *Please address the final points indicated by reviewer 3*

a) *In Materials & Methods, Cell lines: The method used for testing mycoplasma contamination is missing.*

This has been attended to.

b) *Legend Fig. 9: The authors should mention that samples patient 1 to 4 are from pleural effusions and patient 5 - 7 from ascites.*

This has now been clarified in the revised text.

c) *New data shown in Fig. 8D: The authors should mention and comment on the surprising difference between the results for EnAd-SA-EpCAMBITE versus EnAd-CMV-EpCAMBITE.*

A possible explanation for the finding that the SA virus did not induce an increase in CD3+ cells at this time-point is that BiTE expression from the SA promoter will occur later in the infection cycle only when the MLP is activated in cells that are permissive to virus replication (See Figure 5B and Figure EV4). Proliferation is the last functional readout to manifest following activation, which may explain why the SA is able to induce significant cytotoxicity and CD25 upregulation, despite no observable increase in CD3 numbers. This is now discussed further in the revised text.

2) *Movies cannot be referenced from EV figures. Please rename them as Movie EV1, Movie EV2, etc. Also, the legends should be removed from the main manuscript and zipped together with the movie file. Please amend EV figure legends after removal of movie links. Finally, please make sure the callouts in the manuscript are updated.*

This has now been attended to.

3) *Please remove the Fig1S legend from the manuscript and combine with Fig. S1 I the appendix. Please use Appendix Figure S1 nomenclature.*

This has now been corrected.

4) *Please update references to the 20 Names et al. format.*

The references are now in the requested format.

5) *Please remove "Expanded View" prefix from the figure callouts.*

This has now been attended to.

6) *Please correct Fig. EC3F callout I the manuscript*

This has now been corrected.

7) *Please make scale bars more visible (contrasted colour) in Fig. 5E and add to Fig EV4A*

This has now been attended to.

8) *Please remove formatting tracking information, as it is no longer needed at this stage.*

This has been attended to.

- 9) *We encourage the publication of source data, with the aim of making primary data more accessible and transparent to the reader. Would you be willing to provide a PDF file per figure that contains the original, uncropped and unprocessed scans of all or at least the key gels used in the manuscript and/or source data sets for relevant graphs? The files should be labeled with the appropriate figure/panel number, and in the case of gels, should have molecular weight markers; further annotation may be useful but is not essential. The files will be published online with the article as supplementary "Source Data" files. If you have any questions regarding this just contact me.*

Source data has been included for each figure where possible.